# PREDNEXT: EXPLICIT CROSS-VIEW TEMPORAL PREDICTION FOR UNSUPERVISED LEARNING IN SPIKING NEURAL NETWORKS

**Yiting Dong**[1,2]**, Jianhao Ding**[1,2]**, Zijie Xu**[1,2,3]**, Tong Bu**[1,2,3]**, Zhaofei Yu**[1,2,3]*, **Tiejun Huang**[1,2,3]

School of Computer Science, Peking University[1]
State Key Laboratory of Multimedia Information Processing, Peking University[2]
Institute for Artificial Intelligence, Peking University[3]
`{dongyiting,2506398078,putong30,yuzf12,tjhuang}@pku.edu.cn`
`{zjxu25}@stu.pku.edu.cn`

## ABSTRACT

Spiking Neural Networks (SNNs), with their temporal processing capabilities and biologically plausible dynamics, offer a natural platform for unsupervised representation learning. However, current unsupervised SNNs predominantly employ shallow architectures or localized plasticity rules, limiting their ability to model long-range temporal dependencies and maintain temporal feature consistency. This results in semantically unstable representations, thereby impeding the development of deep unsupervised SNNs for large-scale temporal video data. We propose PredNext, which explicitly models temporal relationships through cross-view future Step Prediction and Clip Prediction. This plug-and-play module seamlessly integrates with diverse self-supervised objectives. We firstly establish standard benchmarks for SNN self-supervised learning on UCF101, HMDB51, and MiniKinetics, which are substantially larger than conventional DVS datasets. PredNext delivers significant performance improvements across different tasks and self-supervised methods. PredNext achieves performance comparable to ImageNet-pretrained supervised weights, through unsupervised training solely on UCF101. Additional experiments demonstrate that PredNext, distinct from forced consistency constraints, substantially improves temporal feature consistency while enhancing network generalization capabilities. This work provides a effective foundation for unsupervised deep SNNs on large-scale temporal video data.

## 1 INTRODUCTION

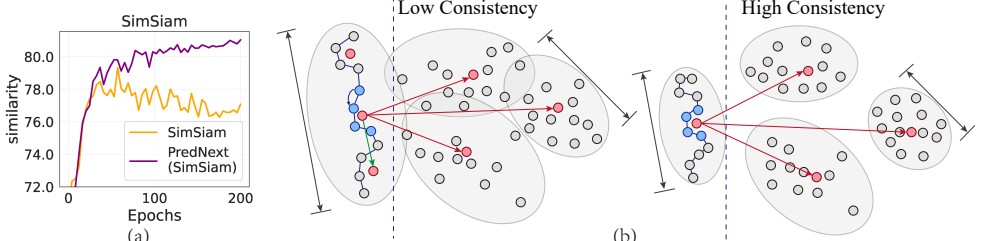

Figure 1: **Analysis of temporal consistency.** (a) Evolution of inter-frame feature similarity during SNN training. (b) Distribution of video features in high-dimensional space, demonstrating more concentrated clustering for high-consistency temporal representations. **Blue** points represent features from different timesteps of the same video, while **red** points indicate cluster centers in nearby feature space locations. **Green** and **red** arrows denote intra-video feature attraction across frames and inter-video feature repulsion respectively.

Unsupervised learning has garnered significant attention in artificial intelligence for its capacity to extract meaningful representations from unlabeled data (Barlow, 1989; Bengio et al., 2012; Liu

---
*Corresponding Author

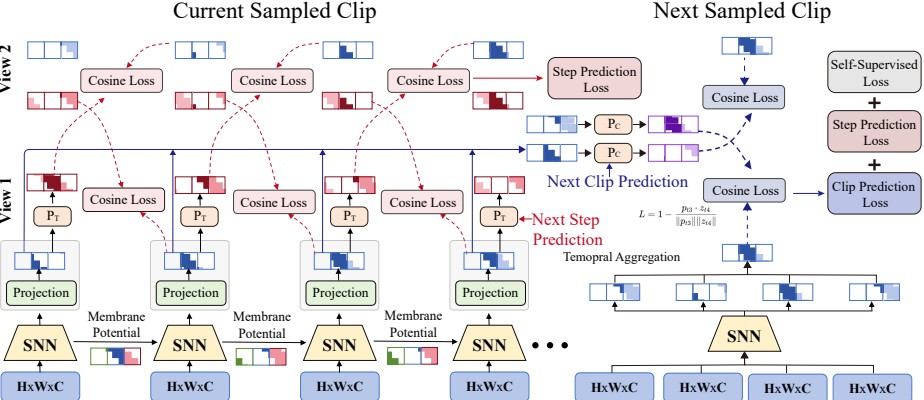

Figure 2: **PredNext algorithmic framework.** PredNext incorporates `Step Prediction` and `Clip Prediction` components for predicting features at the next step and in subsequent sampled clips from the same video, respectively. As an auxiliary module, PredNext can be seamlessly integrated into existing self-supervised learning methods. **Red arrows** indicate the Step Prediction pathway, while **Blue arrows** denote the Clip Prediction pathway.

et al., 2021), substantially reducing dependence on extensive manual annotation. By revealing inherent structures and patterns in unlabeled data, this approach more accurately reflects natural human learning processes (Hinton & Sejnowski, 1999; Chen et al., 2020; He et al., 2020). Spiking neural networks (SNNs), with their characteristics of simulating brain functioning principles (Maass, 1997; Diehl & Cook, 2015; Wu et al., 2018), constitute an ideal framework for unsupervised learning research (Gerstner & Kistler, 2002; Tavanaei et al., 2019). Nevertheless, current research on unsupervised learning in SNNs has primarily concentrated on shallow architectures or synaptic plasticity-based methods (Diehl & Cook, 2015; Kheradpisheh et al., 2018; Dong et al., 2023). The challenges in extending these approaches to deep architectures, particularly when processing complex temporal data, predominantly arise from the limited capacity of current deep SNN models to effectively capture and leverage long-term temporal dependencies (Wu et al., 2018; Fang et al., 2021b). Efficient processing of large-scale, temporally rich data, especially video, is essential for developing robust unsupervised learning systems capable of generating richer, more semantically meaningful feature representations for downstream applications.

The temporal processing capability of spiking neural networks stems from the intrinsic dynamics of spiking neurons, which serve as information carriers across timesteps (Zenke & Vogels, 2021; Neftci et al., 2019). Standard LIF neurons accumulate membrane potential to retain temporal information and emit discrete spikes when the potential exceeds a threshold. However, this elementary integrate-and-fire mechanism proves inadequate for processing large-scale video data with complex temporal dependencies. Additionally, Unlike ANNs employing temporal downsampling (Tran et al., 2015; Carreira & Zisserman, 2017), SNNs typically preserve original temporal resolution, potentially resulting in feature instability without appropriate temporal aggregation. Consequently, we suggest that intrinsic neuronal dynamics alone are insufficient for complex temporal information processing, necessitating the integration of explicit temporal modeling mechanisms to enhance the temporal processing capabilities of SNNs.

Furthermore, we argue that effective temporal modeling should enhance consistency among features extracted across different timesteps. To illustrate this point, Figure 1(a) illustrates the evolution of feature consistency on UCF101(Soomro et al., 2012) as training progresses. The results demonstrate that as models converge, semantic extraction capability im-

Table 1: Summary of commonly used DVS and video datasets.

| #dataset | #classes | #object | #temporal | #scale |
|---|---|---|---|---|
| DVS-Guesture | $1.3K \times 10s$ | action | Real Scene | Small |
| CIFAR10-DVS | $10K \times 1.2s$ | images | Camera Shift | Small |
| N-Caltech101 | $9K \times 0.3s$ | images | Camera Shift | Small |
| **UCF101** | $13K \times 4s$ | action | Real Scene | **Medium** |
| **HMDB51** | $6.7K \times 7s$ | action | Real Scene | **Medium** |
| **miniKinetics** | $80K \times 10s$ | action | Real Scene | **Large** |

proves significantly while feature distributions across timesteps become increasingly consistent. Ideally, as shown in Figure 1(b), high-consistency SNNs should extract stable high-level semantic features (action types, object categories) that remain invariant to temporal fluctuations(Pan et al., 2021; Han et al., 2020b). While directly constraining temporal consistency might seem intuitive, however, our experiments reveal that such enforced consistency constraints actually impair performance.

Based on the preceding analysis, we propose ***PredNext***, that explicitly models temporal relationships and enhances feature consistency in unsupervised spiking neural networks by predicting future features across contrastive views. As illustrated in Figure 2, PredNext operates as a plug-and-play module that seamlessly integrates with existing self-supervised learning algorithms. The framework comprises two complementary mechanisms: Step Prediction, which predicts representations at subsequent timesteps, and Clip Prediction, which predicts features from future temporal clips, while cross-view prediction enhances feature discrimination. PredNext is based on the hypothesis that by explicitly modeling temporal relationships both within and between clips, features with higher semantic density should better predict future representations while excluding low-level dynamic information, thus naturally improving cross-temporal feature consistency.

Due to the scarcity of unsupervised methods for SNNs, we adapted established self-supervised approaches to SNN architectures as benchmarks and reproduced some video unsupervised learning methods. We conducted experiments using UCF101(Soomro et al., 2012) and MiniKinetics(Carreira & Zisserman, 2017) for pre-training, which offer greater scale and richer temporal dependencies than conventional DVS datasets(Li et al., 2017; Orchard et al., 2015)(as shown in Table 1). Results demonstrate that PredNext yields significant performance gains across self-supervised methods while substantially enhancing temporal consistency of extracted features. Our empirical study confirms that superior feature extraction capability corresponds to higher temporal feature consistency, while forcibly imposing consistency constraints degrades performance. Furthermore, experiments show that SNNs, like ANNs, benefit from larger-scale datasets in video processing tasks.

## 2 METHODS

### 2.1 SELF-SUPERVISED LEARNING IN SNNS

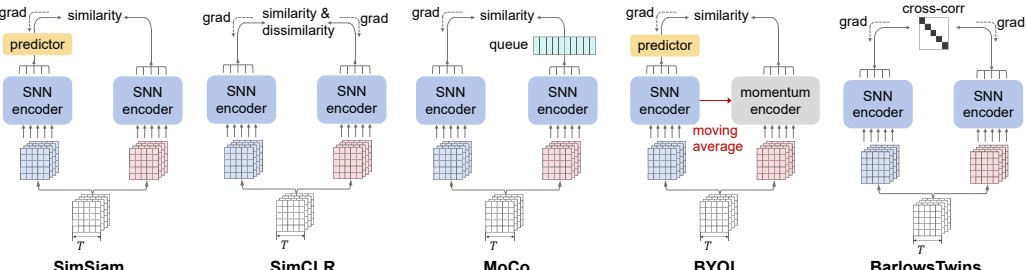

Figure 3: **Implementation for self-supervised learning in SNNs**, encompassing SimCLR, MoCo, SimSiam, BYOL, BarlowTwins. Temporal features are aggregated following SNN encoder.

Given the absence of systematic investigations into self-supervised learning for deep spiking neural networks, we first adapted prevailing self-supervised methods to SNN architectures to establish comparative baselines for our proposed PredNext approach. As depicted in Figure 3, we implemented SNN variants of both contrastive methods (SimCLR(Chen et al., 2020), MoCo(He et al., 2020), BarlowTwins(Zbontar et al., 2021)) and negative-sample-free approaches (SimSiam(Chen & He, 2021), BYOL(Grill et al., 2020)).

Formally, let $x \in D$ denote a clip of length $t$ sampled from dataset $D$. Through data augmentation $H(x)$, we obtain two views $x_i^t$ and $x_j^t$. These views, processed through feature extractors and MLP projection heads, yield representations $z_i^t$ and $z_j^t$. Self-supervised learning aims to minimize distances between representations from different views of the same sample while maximizing distances between representations from different samples. For SNNs, we follow convention by computing the time-averaged representation $z_i = \sum_{t=1}^{T} z_i^t / T$ as the final feature. SimCLR and MoCo implementations utilize the InfoNCE loss function:

$$L = -\log \frac{\exp(sim(z_i, z_j)/\tau)}{\sum_{k=1}^{N} \exp(sim(z_i, z_k)/\tau)} \quad (1)$$

Here, $sim(\cdot, \cdot)$ denotes cosine similarity, $\tau$ represents the temperature parameter, and $N$ is the batch size. SimCLR utilizes in-batch samples as negative examples, whereas MoCo maintains a dynamic

feature queue for negative samples with a momentum encoder. SimSiam and BYOL employ a predictor network $h$ that maps representations between views while minimizing their distance:

$$L = 1 - \frac{z_j}{\|z_j\|_2} \cdot \frac{h(z_i)}{\|h(z_i)\|_2} \qquad (2)$$

where, BYOL employs a momentum encoder for target network updates, while SimSiam utilizes a weight-shared siamese network with stop-gradient operations to prevent collapse. BarlowTwins, conversely, minimizes feature redundancy using the following loss function:

$$L = \sum_i (1 - C_{ii})^2 + \lambda \sum_i \sum_{j \neq i} C_{ij}^2 \qquad (3)$$

where $C$ denotes the cross-correlation matrix of batch-normalized features, and $\lambda$ is the hyperparameter balancing these competing objectives.

Our SNN reproduction for self-supervised learning methods utilizes a SEW ResNet architecture (Fang et al., 2021a) for feature extraction. Across all experiments, we employ the AdamW optimizer (initial learning rate: 2e-3, weight decay: 1e-4) with cosine annealing scheduling and a batch size of $b = 256$. For UCF101 and HMDB51, we use $128 \times 128$ crops with 200 training epochs, with extracted $T = 16$ frames with a stride of $\tau = 2$; for MiniKinetics, $114 \times 114$ crops with 120 epochs. We extract $T = 8$ frames with a stride of $\tau = 8$. Data augmentation follows protocols established in Feichtenhofer et al. (2021). Validation employs 3 clips per video for inference. Comprehensive architectural and hyperparameter details are provided in the appendix.

---

**Algorithm 1** PredNext Training Procedure

---

**Require:** Dataset $D$, data augmentation function $H$, feature extractor and projection head $F$, temporal prediction head $P_T, P_C$, self-supervised loss function $L_{ssl}$, weight coefficient $\alpha$
**Ensure:** Trained feature extractor $F$
 1: **for** each mini-batch **do**
 2:     // Get features from two augmented views
 3:     $x_i = H(x), x_j = H(x)$
 4:     $z_i^t = F(x_i^t), z_j^t = F(x_j^t)$ for $t = 1...T$
 5:     // Compute original self-supervised loss
 6:     $L_{ssl}$ = self-supervised loss based on $z_i$ and $z_j$
 7:     // Compute PredNext predicted features
 8:     $p_i^t = P_T(z_i^t), p_j^t = P_T(z_j^t)$ for $t = 1...T - 1$
 9:     $c_i = P_C(z_i), c_j = P_C(z_j)$
10:     // Compute PredNext loss
11:     $L_{pred} = 0.25 \cdot (\sum_t (Q(p_i^t, z_j^{t+m}) + Q(p_j^t, z_i^{t+m})) + M(c_i, z_j^*) + M(c_j, z_i^*))$
12:     // Compute total loss and update parameters
13:     $L = (1 - \alpha) \cdot L_{ssl} + \alpha \cdot L_{pred}$
14:     Update parameters of $F$ and $P_T, P_C$ to minimize $L$
15: **end for**

---

## 2.2 PREDNEXT

PredNext serves as a plug-and-play auxiliary module seamlessly integrable with diverse self-supervised learning frameworks. As depicted in Figure 2, it introduces temporal prediction as an auxiliary objective while preserving the original self-supervised paradigm. Inspired by Predictive Coding theory(Huang & Rao, 2011; Spratling, 2017), PredNext explicitly models temporal relationships through future representation prediction. This approach operates on the principle that semantically rich features should accurately predict their next semantical feature, whereas features capturing only low-level dynamics cannot generate effective predictions.

PredNext comprises three main components: an SNN feature extractor and a nonlinear MLP projection head (jointly denoted as F), alongside two temporal prediction heads ($P_T$ and $P_C$) for next-timestep and next-clip predictions. The Step Predictor $P_T$ establishes mappings between current and future timestep features, while the Clip Predictor $P_C$ models relationships between current and future clip representations. Both predictors employ two-layer MLPs with dimensions matching the

projection head output. For augmented clips $x_i^t$ and $x_j^t$, we obtain representations $z_i^t = F(x_i^t)$ and $z_j^t = F(x_j^t)$ that serve both the original self-supervised objective and generating predictions through $p_i^t = P_T(F(x_i^t))$, $p_j^t = P_T(F(x_j^t))$ and $c_i = P_C(\frac{1}{T}\sum_t F(x_i^t))$, $c_j = P_C(\frac{1}{T}\sum_t F(x_j^t))$. Step Predictor's loss function minimizes the divergence between current features and cross-view future features:

$$Q(p_i^t, z_j^{t+m}) = -\sum_t \frac{p_i^t}{|p_i^t|} \cdot \frac{z_j^{t+m}}{|z_j^{t+m}|} \tag{4}$$

where $m$ denotes the prediction time step interval. While Clip Predictor's loss function is defined as:

$$M(c_i, z_j^*) = -\frac{c_i}{|c_i|} \cdot \frac{z_j^*}{|z_j^*|} \tag{5}$$

Where $z_i^*$ and $z_j^*$ denote temporally aggregated features of the subsequently sampled clip. To enhance learning effectiveness, we employ a symmetric design, with the final loss function:

$$L_{pred} = \sum_t (\frac{1}{2}Q(p_i^t, z_j^{t+m}) + \frac{1}{2}Q(p_j^t, z_i^{t+m})) + \frac{1}{2}M(c_i, z_j^*) + \frac{1}{2}M(c_j, z_i^*) \tag{6}$$

We employ cross-view prediction where features from one view ($p_i^t$, $c_i$) predict future features of another view ($z_j^{t+m}$, $z_j^*$), with stop-gradient applied to the target features. This design enhances feature discrimination by requiring the model to disregard view-specific noise. Our ablation studies comparing same-view prediction ($p_i^t$ predicting $z_i^{t+m}$) against cross-view prediction demonstrate that the latter yields superior generalization performance. PredNext's complete training procedure is outlined in Algorithm 1. The final optimization objective combines both learning targets:

$$L = (1 - \alpha) \cdot L_{ssl} + \alpha \cdot L_{pred} \tag{7}$$

Where weight coefficient $\alpha$ balances their relative importance.

**Base settings:** As PredNext is model-agnostic and functions as a plug-and-play component across methods, we standardized its parameters throughout our experiments. Following SimSiam (Chen & He, 2021), the temporal prediction head $P_T$ and $P_C$ comprises a 2-layer MLP with batch normalization, using a 128-dimensional hidden layer while maintaining output dimensions consistent with $F(x)$'s feature representation.

**Comparison with Predictive Coding Methods:**

Predictive coding approaches have attracted considerable research interest, particularly for temporal data processing. DPC/MemDPC(Han et al., 2019; 2020a) implement dense predictions on video sequences and utilize dedicated temporal aggregator networks to process intermediate temporal variables. Lorre et al.(Lorre et al., 2020) developed a CPC-like

Table 2: Summary of commonly used DVS and video datasets.

| methods | no additional module needed | step pred | clip pred |
|---|---|---|---|
| DPC | ✗ | ✓ | ✗ |
| memDPC | ✗ | ✓ | ✗ |
| CPC-like(Lorre's) | ✗ | ✓ | ✗ |
| PrredNext | ✓ | ✓ | ✓ |

approach for future timestep feature prediction. As shown in Table 2, in contrast, PredNext employs cross-view prediction with a more streamlined architecture that eliminates the need for complex auxiliary structures, functioning as a modular component integrable with existing methodologies.

## 3 EXPERIMENTS

### 3.1 DATASET AND IMPLEMENTATION

**Datasets details** In contrast to traditional DVS datasets, unsupervised learning paradigms necessitate large-scale datasets to extract meaningful representations. UCF101(Soomro et al., 2012) and HMDB51(Kuehne et al., 2011) are medium-scale video benchmarks widely adopted in action recognition research. UCF101 encompasses $13,320$ video clips across 101 action classes, while HMDB51 contains $6,766$ clips with 51 classes. miniKinetics(Carreira & Zisserman, 2017; Xie et al., 2018), an common subset of Kinetics-400, includes 200 classes with about 400 training and 25 validation instances per class, maintaining diversity and complexity while reducing computational requirements.

**Implementation details** To ensure experimental rigor and comparative validity, we maintain configurations of PredNext aligned with established baselines. We employ SEW ResNet (Fang et al.,

Table 3: **Comparative results after fine-tuning under different self-supervised methods.** *Top-1* and *Top-5* accuracies are reported. Models were trained using various pre-training datasets and evaluated on different fine-tuning datasets. * indicates results reproduced according to our setup.

| | finetune datasets | ucf101 | | hmdb51 | | miniKinetics | |
|---|---|---|---|---|---|---|---|
| **method** | **Initial weights** | top-1 | top-5 | top-1 | top-5 | top-1 | top-5 |
| Supervised | random init | 44.07 | 70.84 | 18.04 | 45.69 | 40.53 | 68.59 |
| Supervised | ImageNet init | 64.42 | 87.36 | 34.31 | 67.84 | 50.48 | 76.53 |
| Supervised | ImageNet + miniKinetics init | 70.02 | 91.62 | 44.97 | 78.37 | - | - |

| **pre-train** | **ucf101** | | | | **miniKinetics** | | | | |
|---|---|---|---|---|---|---|---|---|---|---|
| **finetune** | **ucf101** | | **hmdb51** | | **ucf101** | | **hmdb51** | | **miniKinetics** | |
| **method (Initial weights)** | top-1 | top-5 | top-1 | top-5 | top-1 | top-5 | top-1 | top-5 | top-1 | top-5 |
| **SimCLR** | 57.04 | 83.82 | 30.59 | 64.97 | 59.03 | 85.96 | 35.42 | 67.97 | 50.61 | 77.16 |
| **MoCo** | 49.70 | 79.70 | 28.04 | 62.22 | 45.63 | 76.55 | 20.72 | 46.86 | 42.65 | 70.23 |
| **BYOL** | 56.41 | 83.18 | 29.35 | 64.58 | 59.27 | 86.23 | 36.74 | 68.24 | 51.23 | 77.69 |
| **BarlowTwins** | 56.15 | 84.25 | 30.33 | 64.12 | 58.04 | 85.83 | 36.53 | 68.17 | 51.28 | 77.61 |
| **SimSiam** | 50.81 | 81.07 | 28.10 | 63.46 | 43.77 | 74.89 | 19.08 | 45.75 | 41.52 | 69.75 |
| **SimSiam (ImageNet)** | 70.32 | 91.56 | 39.65 | 74.35 | 68.70 | 91.91 | 36.67 | 73.53 | - | - |
| $\rho$**SimSiam**$(\rho = 1)$ | 52.05* | 81.75* | 28.56* | 64.30* | - | - | - | - | - | - |
| **CVRL(SimSiam-based)** | 52.81* | 82.15* | 29.22* | 64.38* | - | - | - | - | - | - |
| **PredNext**$_{SimCLR}$ | 59.47 | 85.28 | 31.58 | 66.19 | 61.06 | 87.21 | 36.80 | 68.37 | 53.61 | 78.59 |
| **PredNext**$_{MoCo}$ | 54.98 | 82.87 | 29.60 | 64.31 | 51.60 | 79.65 | 25.69 | 51.37 | 46.51 | 73.64 |
| **PredNext**$_{BYOL}$ | 58.58 | 83.82 | 31.57 | 64.51 | 62.01 | 88.26 | 37.25 | 69.28 | 54.37 | 79.61 |
| **PredNext**$_{BarlowTwins}$ | 59.76 | 84.85 | 31.18 | 66.01 | 62.75 | 88.66 | 37.65 | 69.35 | 54.68 | 79.85 |
| **PredNext**$_{SimSiam}$ | 54.93 | 82.77 | 30.00 | 64.37 | 50.65 | 79.01 | 25.03 | 51.04 | 46.31 | 73.68 |
| **PredNext**$_{SimSiam}$ **(ImageNet)** | 72.24 | 91.81 | 41.50 | 75.42 | 71.66 | 92.07 | 38.63 | 74.25 | - | - |

2021a) as the feature extraction backbone across all experimental conditions. For UCF101 and HMDB51, we crop video frames at $128 \times 128$ resolution, sampling 16 frames with a stride of 2. MiniKinetics processing utilizes $112 \times 112$ resolution with 8 frames with a stride of 8. During evaluation, we perform inference on 3 uniformly sampled clips per test video. Optimizer hyperparameters remain consistent with baseline model configurations. More experimental parameters details are included in the appendix. While optical flow typically enhances performance in video understanding tasks(Han et al., 2020b; Carreira & Zisserman, 2017), we exclude this modality as our investigation primarily focuses on temporal feature consistency in SNNs under unsupervised learning paradigms, and we reserve multimodal integration for subsequent research endeavors.

## 3.2 RESULTS OF UNSUPERVISED REPRESENTATION EVALUATION

We first evaluated the performance of various self-supervised learning methods in baseline spiking neural network implementations, then incorporating PredNext as an auxiliary module to quantify performance enhancements. Following the experimental protocol established in (Han et al., 2019), we utilized UCF101 and MiniKinetics as pre-training datasets and report performance after fine-tuning on different target datasets.

Table 3 presents performance across pretraining and fine-tuning configurations. Even basic SNN self-supervised methods achieve substantial results on action recognition tasks. PredNext consistently yields significant improvements across all methods, demonstrating its effectiveness in enhancing temporal representation learning. Notably, PredNext achieves performance comparable to ImageNet-pretrained supervised weights, through unsupervised training solely on UCF101. Moreover, models trained on larger pretraining datasets consistently show superior performance, confirming that SNNs, like ANNs, benefit significantly from data scale (without MoCo, SimSiam). Interestingly, even trained with same datasets, unsupervised models outperformed those trained with supervision (SimSiam on UCF101), highlighting the research significance of video unsupervised learning in providing stronger generalization. Furthermore, larger datasets provide more effective parameter initialization—models initialized with ImageNet weights and pre-trained solely on UCF101 achieve performance (SimSiam(ImageNet) on UCF101) comparable to supervised learning on MiniKinetics.

We observe that SimSiam and MoCo exhibit relatively lower performance compared to the other three methods. We attribute this to the following reasons: SimSiam lacks negative samples compared to other approaches, leading to relatively unstable training, whereas BYOL enhances stability

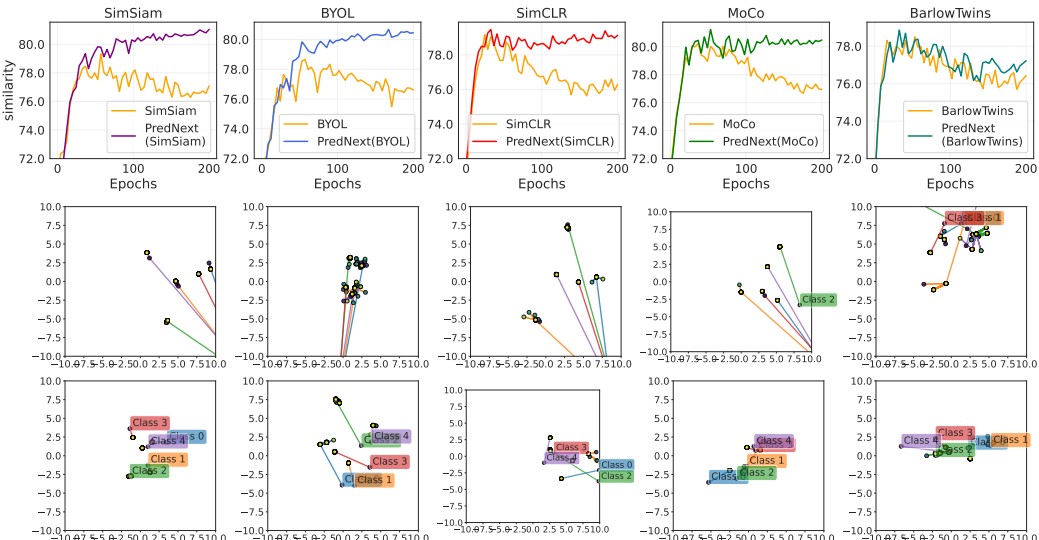

Figure 4: **Analysis of temporal feature visualization. Top row:** evolution of temporal consistency error during training across methods. **Middle** and **bottom rows:** UMAP visualizations of video features from baseline self-supervised methods and their PredNext-enhanced variants, respectively.

through a momentum encoder. On the other hand, MoCo requires maintaining a memory bank as a negative sample repository, which proves challenging for datasets like UCF101 to sustain a large and consistent bank for effective training.

## 3.3 CONSISTENCY CURVES AND MANIFOLD

To examine PredNext's influence on SNN temporal feature representations, we analyzed feature consistency across methods. Figure 4 illustrates the evolution of feature consistency during training. We define feature consistency error as the average cosine distance between representations from different time steps of the same video:

$$E_{consistency} = \frac{1}{N} \frac{1}{T(T-1)} \sum_{i=1}^{N} \sum_{t=1}^{T} \sum_{s=1,s\neq t}^{T} \left(1 - \cos(f_i^t, f_i^s)\right) \qquad (8)$$

where $f_i^t$ represents video $i$'s feature at time $t$, $N$ denotes the sample count, and $T$ indicates time steps per video. Lower values indicate lower temporal feature consistency.

**Consistency Visualization** As Figure 4(top row) demonstrates, consistency errors decrease during training across all methods, indicating progressive learning of stable temporal features before eventual saturation or deterioration. Methods incorporating PredNext maintain comparable early-stage convergence rates to baselines but avoid the post-saturation decline, ultimately achieving significantly lower consistency errors. This confirms our hypothesis that explicit temporal prediction modeling guides networks toward semantically richer, temporally consistent representations.

Table 4: **Comparative results of forced consistency constraint experiments.** $\beta$ denotes constraint intensity; error represents temporal feature consistency deviation.

| UCF101 | SimSiam (ImageNet) | SimSiam $PredNext$ (ImageNet) | Forced Consistency | | |
|---|---|---|---|---|---|
| $\beta$ | - | - | 0.1 | 0.5 | 0.8 |
| top-1 | 70.32 | $72.24_{+1.92}$ | $70.45_{+0.13}$ | $65.69_{-4.63}$ | $60.35_{-9.97}$ |
| consistency | 0.773 | $0.819_{+0.046}$ | $0.803_{+0.03}$ | $0.852_{+0.08}$ | $0.884_{+0.11}$ |

To further visualize learned representations, we applied UMAP (McInnes et al., 2018) for dimensionality reduction on test set samples, as shown in Figure 4 (middle and bottom rows). Original self-supervised methods generate temporally dispersed features, with representations from different time steps often widely separated. In contrast, PredNext-enhanced methods significantly improve feature clustering, with same-video feature points exhibiting substantially tighter grouping.

Table 5: **Video retrieval performance comparison.** R@1, 5, 10, 20 denote recall rates at corresponding rank thresholds. Evaluations performed on UCF101 and HMDB51 datasets. All models pretrained on UCF101 split 1.

| UCF101 pretrain | UCF101 | | | | HMDB51 | | | |
|---|---|---|---|---|---|---|---|---|
| methods | R@1 | R@5 | R@10 | R@20 | R@1 | R@5 | R@10 | R@20 |
| SimCLR | 34.58 | 55.72 | 65.50 | 74.70 | 12.22 | 34.71 | 49.67 | 64.71 |
| SimCLR$_{PredNext}$ | 37.09 | 56.01 | 66.38 | 75.20 | 13.60 | 35.36 | 50.32 | 66.86 |
| SimSiam | 27.84 | 48.53 | 59.79 | 71.56 | 11.70 | 32.68 | 45.95 | 60.98 |
| SimSiam$_{PredNext}$ | 36.27 | 55.70 | 65.13 | 74.15 | 13.20 | 35.16 | 47.32 | 64.05 |
| SimSiam$_{PredNext}$ (ImageNet) | 53.19 | 69.39 | 76.53 | 83.11 | 15.95 | 40.46 | 53.53 | 68.43 |

**Forced Consistency Constraints** Furthermore, we conducted a control experiment with forced consistency constraints by directly adding an explicit constraint to the loss function, compelling feature similarity across different time steps of the same video:

$$L_{forced} = L_{ssl} + \beta \cdot \mathbb{E}_{i,t,s}[1 - \cos(f_i^t, f_i^s)] \tag{9}$$

This approach diverges from PredNext by eliminating prediction heads and prediction processing. As shown in Table 4, this direct constraint indeed rapidly reduces consistency errors, even faster than PredNext. However, analysis of the relationship between feature consistency and downstream task performance reveals that despite generating more consistent features, forced constraints yield inferior fine-tuning performance compared to PredNext's representations.

Therefore, these findings demonstrate that superior feature extraction capability corresponds with higher temporal feature consistency and stability. However, simply enforcing consistency through constraints does not necessarily lead to better feature extraction capabilities. High-quality features capture semantic information in videos (such as action types, object categories), which should naturally remain relatively stable over time periods. Forced consistency constraints potentially suppress critical temporal dynamics, yielding oversimplified representations with low discriminative capacity.

## 3.4 VIDEO RETRIEVAL

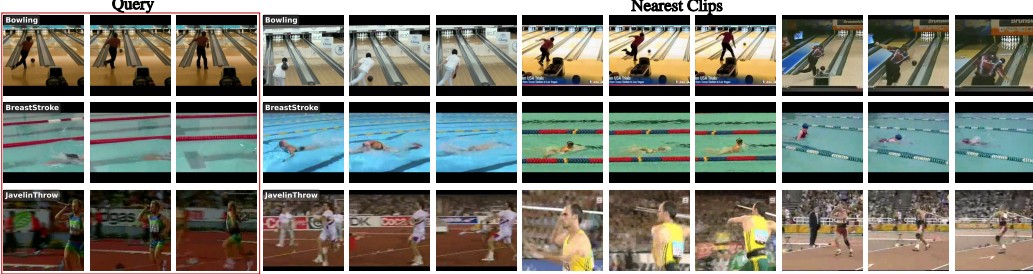

Figure 5: **Visualization of retrieval results.** Query videos (in left) with corresponding Top-3 retrieval results. Results for three query samples shown, with one sample per row.

**Retrieval Results** To further evaluate the semantic representation capabilities, we conducted video retrieval evaluations following (Han et al., 2019). Using UCF101's split 1 validation set as queries and the corresponding training split as retrieval candidates, we uniformly sampled 10 frames per video and extracted temporally aggregated features from pretrained models. The retrieval process employed a Nearest Neighbor(NN) search. we identified the K closest videos to each query and calculated category matching performance (Recall@K). Table 5 presents video retrieval performance across self-supervised methods using Recall@1,5,10,20 metrics. Results demonstrate that PredNext integration yields significant improvements across all retrieval benchmarks, confirming its capacity to facilitate more precise semantic representations.

**NN Visualization** Figure 5 provides visualization examples retrieval from PredNext's features. Query (Figure 5 (left)) videos with their corresponding Top-3 retrieval results (Figure 5 (right)) illustrate that PredNext can retrieve semantically consistent videos despite significant visual variations in varied camera angles, player appearances, and visual contexts.

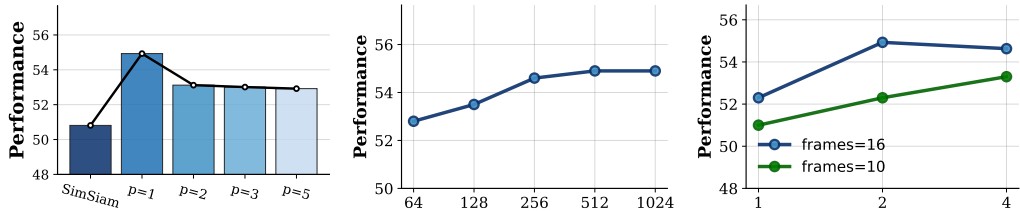

Figure 6: (a) Impact of prediction step length on model performance. (b) Influence of prediction head hidden layer dimensionality on model efficacy. (c) Effects of temporal length and sampling rate on model performance.

Table 6: **Ablation studies.** Performance comparison following removal of step prediction and clip prediction components. Experiments conducted on SimSiam and SimCLR. All models pretrained on UCF101 split1.

| | | SimSiam | | | | SimCLR | | | |
|---|---|---|---|---|---|---|---|---|---|
| **step** | **clip** | **ucf101** | | **hmdb51** | | **ucf101** | | **hmdb51** | |
| **prediction** | **prediction** | top-1 | top-5 | top-1 | top-5 | top-1 | top-5 | top-1 | top-5 |
| ✗ | ✗ | 50.81 | 81.07 | 28.10 | 63.46 | 57.04 | 83.82 | 30.59 | 64.97 |
| ✓ | ✗ | 51.33 | 81.26 | 28.76 | 63.73 | 57.86 | 84.14 | 30.65 | 65.03 |
| ✗ | ✓ | 54.40 | 82.37 | 29.54 | 64.12 | 59.21 | 84.96 | 31.18 | 66.01 |
| ✓ | ✓ | **54.93** | **82.77** | **30.00** | **64.37** | **59.48** | **85.28** | **31.57** | **66.34** |

## 4 ABLATION STUDIES

**Impact of Prediction Head** $P_T, P_C$ Table 6 illustrates the impact of Prediction Heads $P_T$ and $P_C$ on model performance. Both prediction components independently enhance performance, while their combination in PredNext yields further improvements. Clip prediction demonstrates more substantial effects than step prediction, which we attribute to its coverage of temporal information across a longer time range, facilitating acquisition of richer temporal representations.

**Impact of Prediction Step Length** Prediction step length determines the temporal distance for feature prediction. Figure 6(a) illustrates performance across varying step lengths. Optimal performance typically occurs at step length 1, with declining performance at longer intervals. We analyze that when $m > 1$, adjacent timesteps lose the ability to interact for prediction, as larger $m$ values cause the model to skip nearby temporal moments, resulting in significantly sparser predictive interactions compared to $m = 1$ and consequently leading to performance degradation.

**Impact of Cross-view Prediction** Table 7 compares four prediction strategies: cross-view prediction, same-view prediction, and their standalone implementations without original self-supervised objectives. Cross-view prediction consistently outperforms alternatives across all methods. By predicting features across different augmentations, models must isolate semantically meaningful features, while same-view-only prediction leads to representation collapse.

**Impact of Prediction Head Size** Figure 6 (b) illustrates how prediction head $P_T, P_C$ hidden dimensionality affects model performance. Testing dimensions from 64 to 1024 reveals that performance improves with increasing dimensionality but stabilizes beyond 256 dimensions. This indicates that the prediction head re-

Table 7: **Comparative results between same-view and cross-view prediction.** "only" indicates training without original self-supervised objectives.

| dataset | cross-view | same-view | cross-view only | same-view only |
|---|---|---|---|---|
| UCF101 | **54.93** | $53.66_{-1.27}$ | $52.37_{-2.56}$ | $5.03_{-49.90}$ |
| HMDB51 | **30.00** | $29.67_{-0.33}$ | $29.41_{-0.59}$ | $3.07_{-26.93}$ |

quires sufficient representational capacity for effective temporal modeling but becomes parameter-inefficient beyond certain thresholds. We selected 512 dimensions as the optimal configuration, balancing performance with computational efficiency. Notably, the prediction head introduces minimal additional parameters compared to the feature extraction backbone and is utilized exclusively during training, introducing no computational overhead during inference.

**Impact of Time Lengths and Sampling Stride** Figure 6 (c) illustrates how clip length and sampling stride influence model performance. Evaluating combinations of sequence lengths (10, 16 frames) and sampling intervals (1, 2, 4) reveals consistent performance improvements with both increased sequence length and wider sampling intervals. This pattern suggests that sequences span-

Table 9: **Comparison with other SNN/ANN methods on UCF101.** * denotes stronger data augmentation. **pretrain** indicates whether ImageNet pretraining is used. Due to varying model capacities, the effectiveness of ImageNet pretrained weights differs. **ANN** indicates ANN-based models. † indicates results reported by unofficial split and size.

| method | Un-/Sup | model | pretrain | pretrain Acc in ImageNet | Top1 | Top5 |
|---|---|---|---|---|---|---|
| vanilla | supervised | ResNet 18(ANN) | ✗ | - | 40.7 | 63.8 |
| vanilla | supervised* | ResNet 18(ANN) | ✗ | - | 53.2 | 78.3 |
| vanilla | supervised* | ResNet 34(ANN) | ✗ | - | 54.2 | 77.4 |
| vanilla | supervised* | ResNet 50(ANN) | ✗ | - | 54.3 | 77.5 |
| **ReSpike** (Xiao et al. (2025)) | supervised | ResNet 18(ANN) +MS-ResNet18 | ✓ | 73.2 | 77.5 | 93.9 |
| **SVFormer-st** (Yu et al. (2024)) | supervised* | SVFormer-st | ✓ | 82.9 | 80.2 | - |
| **LSM+STDP** (Panda & Srinivasa (2018)) | hand-crafted +supervised | LSM-16.2M | - | - | 70.2† | - |
| STS ResNet (Samadzadeh et al. (2023)) | supervised | STS ResNet | ✗ | - | 42.1 | - |
| **SimSiam** | **unsupervised** | ResNet 18(ANN) | ✗ | - | 49.3 | 78.6 |
| **SimSiam**$_{PredNext}$ | **unsupervised** | SEW ResNet18 | ✗ | - | 54.9 | 82.8 |
| **SimCLR**$_{PredNext}$ | **unsupervised** | SEW ResNet18 | ✗ | - | 59.5 | 85.3 |
| **SimSiam**$_{PredNext}$ | **unsupervised** | SEW ResNet18 | ✓ | 63.2 | 72.2 | 91.8 |
| **SimSiam**$_{PredNext}$ | **unsupervised** | SEW ResNet34 | ✓ | 67.0 | 74.1 | 93.1 |
| **SimSiam**$_{PredNext}$ | **unsupervised** | SEW ResNet50 | ✓ | 67.8 | 74.2 | 93.1 |

ning broader temporal ranges provide richer contextual information, enabling more comprehensive semantic understanding of actions.

**Impact of weighting coefficient** $\alpha$ Since our method jointly optimizes the original self-supervised loss and prediction loss, we investigate the impact of varying weighting coefficients $\alpha$. Table 8 presents the performance of

Table 8: **Comparative results between different weight coefficient** $\alpha$, where $\alpha = 0$ corresponds to the original SSL method and $\alpha = 1$ equals the cross-view only setting.

| methods | dataset | 0 | 0.2 | 0.4 | 0.5 | 0.6 | 0.8 | 1.0 |
|---|---|---|---|---|---|---|---|---|
| **SimSiam**$_{PredNext}$ | UCF101 | 50.8 | 52.4 | 53.4 | **54.9** | 53.8 | 52.2 | 52.4 |
| **SimCLR**$_{PredNext}$ | UCF101 | 57.0 | 57.4 | 57.9 | **59.5** | 58.6 | 55.5 | 52.4 |
| **SimSiam**$_{PredNext}$ | HMDB51 | 28.1 | 28.3 | 28.9 | **30.0** | 29.4 | 29.5 | 29.4 |

PredNext on UCF101 and HMDB51 under different $\alpha$ values. Results shows increasing the prediction loss weight($\alpha = 0.5$) yields significant performance improvements, excessively high prediction weights($\alpha = 0.8$), result in performance degradation, ultimately converging to the cross-view only setting at $\alpha = 1$. Complete reliance on the prediction task may overlook important information from the original self-supervised task. Given that $\alpha = 0.5$ consistently exhibits superior performance across all experiments, we adopt this value as the default setting throughout our study.

**Comparison with other SNN/ANN methods** We compare PredNext with other SNN/ANN methods reporting results on UCF101. Table 9 presents the performance of different methods. We observe that model performance correlates significantly with pretrained weight effectiveness. Without ImageNet pretraining, PredNext even outperforms ANN-based supervised baselines (we find that weak augmentation causes ANN collapse on UCF101, thus we employ stronger augmentation than reported basic setting). With ImageNet pretraining, PredNext performs lower compared to methods with larger parameters and ANN supervision, which we contribute to SNN pretrained weights achieving lower performance (63.2% vs. 73.2%). Meanwhile, PredNext performance scales with model size, improving from SEW-ResNet18 to ResNet34. However, on SEW-ResNet50, marginal pretrained weight quality differences prevent further leveraging parameter scale advantages. Notably, PredNext without pretraining weight surpasses self-supervised ANN methods with identical architecture (ResNet-18), demonstrating advantages in SNN self-supervised learning performance.

## 5 CONCLUSION

We present PredNext, a method enhancing unsupervised spiking neural networks through future feature prediction that strengthens temporal consistency. Experimental evidence demonstrates that PredNext delivers significant performance improvements over unsupervised SNN methods while substantially enhancing temporal coherence in network representations.

ETHICS STATEMENT

Our paper does not involve any ethical issues. Our methods and experiments adhere to academic ethical standards without involving any sensitive data or privacy concerns.

REPRODUCIBILITY STATEMENT

We provide detailed experimental settings and hyperparameter configurations in the appendix to ensure that other researchers can reproduce our results. We plan to publicly release our code and pretrained models to facilitate further research and applications within the community.

ACKNOWLEDGMENTS

This work is supported by the National Natural Science Foundation of China (62422601), Beijing Municipal Science and Technology Program (Z241100004224004), Beijing Nova Program (20240484703), National Key Laboratory for Multimedia Information Processing, and Beijing Key Laboratory of Brain-inspired Spiking Large Models.

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

## A    LLM USAGE

In this paper, we restricted the use of LLMs solely for language refinement, without employing these models for paper composition , experimental design, or conceptual development. All core scientific contributions were independently developed by the authors without LLM assistance.

## B    RELATED WORK

### B.1    SPIKING NEURAL NETWORKS

Spiking neural networks (SNNs) are novel neural network models that simulate information processing mechanisms in biological neural systems. Unlike traditional artificial neural networks (ANNs), SNNs transmit and process information through discrete spike signals, offering higher biological interpretability and temporal processing capabilities (Maass, 1997; Gerstner & Kistler, 2002; Roy et al., 2019). In recent years, with advances in hardware technology and algorithmic innovations, SNNs have made progress in image recognition, speech processing, and robotic control (Tavanaei et al., 2019; Wu et al., 2018). However, due to their discontinuous nature, SNNs face challenges in training and optimization, particularly evident in complex tasks such as video understanding. Especially in video understanding tasks, SNNs must process substantial temporal information and complex spatial structures, placing higher demands on their temporal feature learning capabilities (Dong et al., 2024; Fang et al., 2021b). Consequently, enhancing SNN performance in video understanding has emerged as a significant research focus.

### B.2    VIDEO UNSUPERVISED LEARNING

Video unsupervised learning aims to learn meaningful temporal and spatial feature representations from unlabeled video data. In recent years, contrastive learning-based methods have achieved significant progress in video unsupervised learning (Han et al., 2019; Ahsan et al., 2019; Feichtenhofer et al., 2021). These approaches optimize models through contrastive loss functions using constructed positive and negative sample pairs, enabling capture of temporal dynamics and spatial structural information in videos. DPC (Han et al., 2019) iteratively predicts future features by inputting each timestep's features into an external temporal processing module. VideoJigsaw (Ahsan et al., 2019) learns temporal information through video block reorganization. CoCLR (Han et al., 2020b) learns video representations by aligning optical flow with video content. Lorre et al. (Lorre et al., 2020) employ CPC-like methods that predict future features. The $\rho$ series models Feichtenhofer et al. (2021) introduce contrastive methods to the video domain with temporal correlation components. VideoMoCo (Pan et al., 2021) learns through adversarial samples using the MoCo method. Additionally, generative models have been widely applied in video unsupervised learning, learning latent video representations by reconstructing video frames or generating future frames (Wei et al., 2022; Wang et al., 2022). However, most existing video unsupervised learning methods are designed primarily for ANNs, leaving the effective application of these methods to SNNs an urgent problem requiring resolution.

### B.3    UNSUPERVISED LEARNING IN SNNS

Research on unsupervised learning in spiking neural networks (SNNs) has been relatively limited, though it has begun attracting attention in recent years (Diehl & Cook, 2015; Dong et al., 2023; Ma et al., 2025). Existing work primarily focuses on implementing unsupervised learning in SNNs through plasticity rules and local learning algorithms (Diehl & Cook, 2015; Dong et al., 2023; Ororbia, 2024; Saunders et al., 2019). For instance, Spike-Timing-Dependent Plasticity (STDP), a learning rule based on biological neuronal plasticity, has been widely applied in unsupervised learning with SNNs (Bi & Poo, 1998). Additionally, some studies have attempted to apply unsupervised learning methods such as contrastive learning to SNNs (Ma et al., 2025; Bahariasl & Kheradpisheh, 2024), or adapt deep methods originally developed for ANNs (Li et al., 2023). Other approaches focus on relationships between events and images (Hagenaars et al., 2021). However, existing research primarily concentrates on shallow networks without a systematic benchmark methodology, while focusing on image data rather than addressing temporal video data processing.

## C    MORE RETRIEVAL VISUALIZATION RESULTS

We provide additional video retrieval visualization examples here. As observed, PredNext successfully retrieves semantically consistent videos even when significant variations exist in camera angles, athlete appearances, and visual environments. Even in instances of retrieval errors, the retrieved results typically maintain some semantic relevance to the query video.

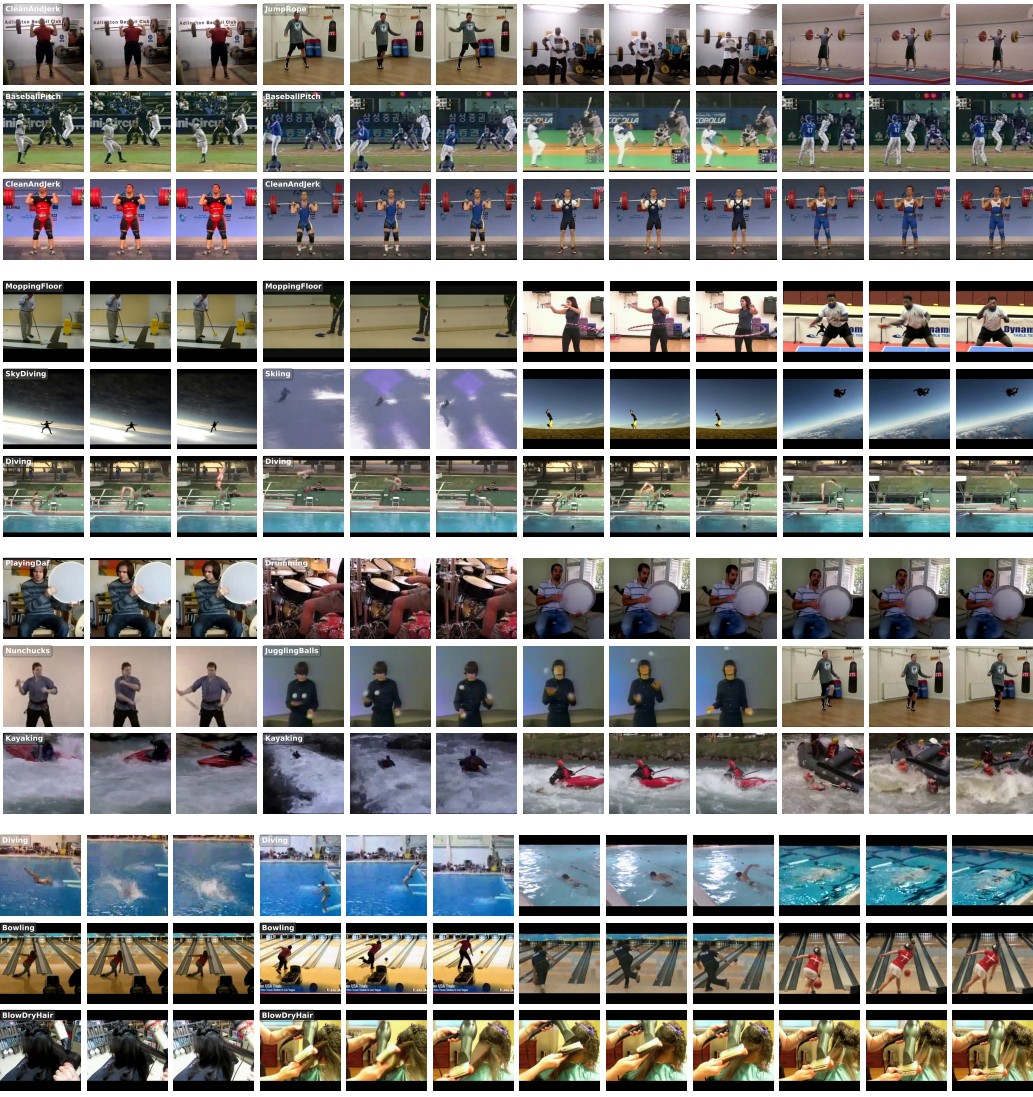

Figure 7: **Visualization of more retrieval results.** Query videos with corresponding Top-3 retrieval results. Results for three query samples shown, with one sample per row.

## D    KNN TRAINING CURVE

To demonstrate the pretraining process, we evaluated the feature representation capability of our models during pretraining using KNN classifiers, which can assess features without downstream task fine-tuning. We conducted evaluations on UCF101 split1. Figure 8 shows the top1/5 accuracy curves of KNN classifiers throughout the pretraining process. As observed, PredNext significantly enhances the model's feature representation capabilities.

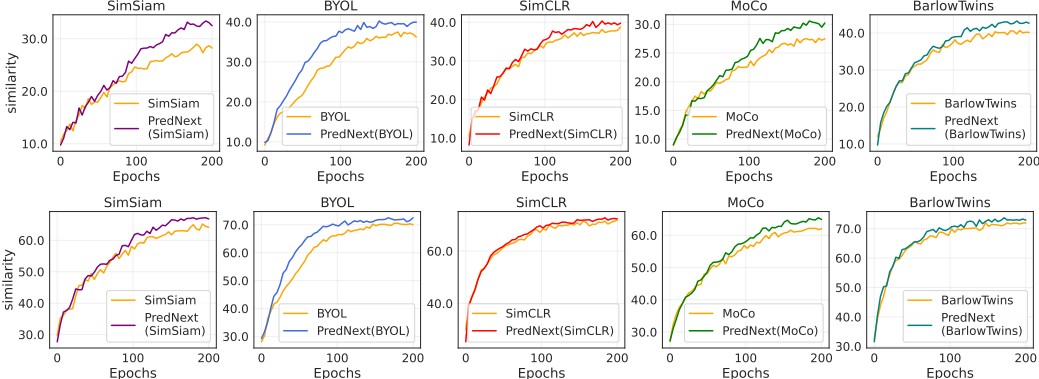

Figure 8: **Visualization of KNN training Curve**, showing Top1 (top) and Top5 (bottom) accuracy curves, respectively.

## E    COMPUTATIONAL ANALYSIS

We provide device resource comparison of PredNext on SimSiam and SimCLR methods in Table 10. PredNext introduces only marginal increases in training time, GPU memory usage, Memory, GPU Memory and FLOPs. This demonstrates that PredNext maintains low computational overhead while improving performance, making it suitable for large-scale training in practical applications.

Table 10: **Computational Analysis of PredNext on SimSiam and SimCLR methods.** $T$ denotes the total number of input frames.

|  | $SimSiam$ | $SimSiam$ $PredNext$ | $SimCLR$ | $SimCLR$ $PredNext$ |
|---|---|---|---|---|
| **GPU devices** | 4 | 4 | 4 | 4 |
| **Training Time** | 1.39min/epoch | 1.43min/epoch | 1.20min/epoch | 1.36min/epoch |
| **GPU Memory** | 12.2G×4 | 12.4G×4 | 12.1G×4 | 12.4G×4 |
| **Memory Peak** | 40GB | 43GB | 37GB | 47GB |
| **FLOPs** | 1.188G×T | 1.193G×T | 1.188G×T | 1.193G×T |
| **Data Workers** | 16 | 16 | 16 | 16 |
| **Throughput** | 114.3frames/s | 111.1frames/s | 132.5frames/s | 116.9frames/s |

## F    DISSCUSSION ON TEMPORAL DYNAMICS IN SNNs

The temporal dynamics in spiking neurons are crucial for the entire network. However, we argue that solely relying on neuronal dynamics to implicitly learn temporal characteristics does not fully exploit the potential of spiking neurons. On one hand, SNN architectures typically borrow from ANN image recognition network designs, which makes networks more prone to spatial bias. Similar observations have been made in ANN-based video models(Goyal et al., 2017; Choi et al., 2019). On the other hand, SNNs lack the progressive temporal aggregation mechanisms present in ANN 3D(Carreira & Zisserman, 2017) convolutional networks, preventing temporal dimensions from undergoing gradual downsampling through pooling layers or larger-stride convolutions as spatial dimensions do, thereby limiting sufficient temporal information extraction. Therefore, we aim to explicitly enhance temporal consistency through architectural design, thereby alleviating the network's spatial bias while improving temporal aggregation capability to more fully leverage the temporal processing capacity of spiking neurons.

## G    THEORETICAL ANALYSIS

In the original method, computation focuses on modeling relationships between sample instances. In this work, we further attend to computational interactions between frames and clips, which are unique characteristics of temporal data.

Video data contains two types of information:

(i) **semantic content** $\mathcal{S}$, such as action categories and object identities, which remains relatively stable over time;

(ii) **low-level noise** $\mathcal{N}$, such as illumination variations and camera shake, whose temporal correlation decays rapidly.

These two information types exhibit fundamentally different temporal correlation characteristics(Taylor et al., 2010; Goyal et al., 2017): semantic content demonstrates long-range correlation $\rho_{\mathcal{S}}(m) \approx e^{-\epsilon_{\mathcal{S}} m}$, while noise exhibits exponential decay $\rho_{\mathcal{N}}(m) \approx e^{-\lambda_{\mathcal{N}} \cdot m}$ Wiskott & Sejnowski (2002), where $\epsilon_{\mathcal{S}} \ll \lambda_{\mathcal{N}}$. This implies that a "sport action" persists across multiple frames, whereas "instantaneous glare at a particular moment" quickly disappears.

PredNext's temporal prediction objective $\mathcal{L}_{\text{pred}}$ is equivalent to maximizing mutual information $I(z^t; z^{t+m})$ or $I(z; z^*)$ . $z^*$ denotes the temporally aggregated representation of next clip. Assuming semantic and noise statistics are approximately independent. This assumption is generally reasonable for video data, as short-term noise and long-term semantics occupy separated signal frequency spectra(Ruderman & Bialek, 1993): $\mathcal{Z} = \rho_{\mathcal{S}}(m) + \rho_{\mathcal{S}}(m)$. For prediction step $m$, the noise mutual information $I(n^t; n^{t+m}) \propto e^{-2\lambda_{\mathcal{N}} m}$ approaches zero, while the semantic mutual information $I(s^t; s^{t+m})$ remains substantial. Consequently, the optimization process naturally prioritizes encoding predictable semantic content while filtering unpredictable noise. Predictability serves as an implicit regularizer that filters out unpredictable noise. This also explains why enforced consistency proves detrimental: the forced constraint $\mathcal{L}_{\text{forced}} = \mathbb{E}_{i,t,s}[1 - \cos(f_i^t, f_i^s)]$ indiscriminately suppresses all temporal variations.

# H SETTING DETAILS

We provide detailed experimental specifications to facilitate the reproduction of our work.

## H.1 EXPERIMENTAL DETAILS

For all experiments, we employed SEW ResNet as the feature extraction backbone network and implemented models using the PyTorch framework. Synchronized batch normalization layers were utilized across all experiments due to multi-GPU training. Automatic mixed precision (AMP) training was employed across all experiments to enhance training efficiency.

**Pre-training** We used the AdamW optimizer with an initial learning rate of **0.002** and weight decay of **1e-6**, applying **cosine annealing** learning rate scheduling. For UCF101, we conducted **200** epochs of training with a **20-epoch** warmup process; for MiniKinetics, **120** epochs with a **12-epoch** warmup. Training utilized mini-batches of size **128**. Data augmentation included random cropping (scale: **(0.2, 0.766)**, ratio: **(0.75, 1.3333)**), horizontal flipping( **p: 0.5** ), color jittering( brightness: **0.6**, contrast: **0.6**, saturation: **0.6**, hue: **0.1**), and random gray( **p: 0.2**). For UCF101, videos were cropped to $128 \times 128$ resolution with **16** frames randomly sampled at a stride of **2**; for MiniKinetics, videos were cropped to $112 \times 112$ resolution with **8** frames randomly sampled at a stride of **8**.

**Fine-tuning** We employed the AdamW optimizer with an initial learning rate of **0.0003** without weight decay, applying cosine annealing scheduling. For UCF101 and HMDB51, videos were cropped to $128 \times 128$ resolution with **16** frames randomly sampled at stride **2**; for MiniKinetics, videos were cropped to $112 \times 112$ resolution with **8** frames randomly sampled at stride **8**. Training used mini-batches of size **128** for **100** epochs on UCF101 and HMDB51, and **50** epochs on MiniKinetics. Evaluation uniformly sampled **3** clips per sample.

## H.2 MODEL DETAILS

For SimCLR, projection layer output dimension was 256 with temperature coefficient 0.5. For MoCo, projection layer output dimension was 256, momentum coefficient 0.99, queue size 4096, and temperature parameter 0.5. For BYOL, projection/prediction layer output dimension was 2048, with prediction layer hidden dimension 512 and momentum coefficient 0.99. For BarlowTwins, projection layer output dimension was 1024. For SimSiam, projection/prediction layer output dimension was 2048, with prediction layer hidden dimension 512. PredNext's prediction heads $P_T$ and $P_C$ both used hidden layer dimension 512, with output dimensions matching the projection layer output dimensions of their respective base self-supervised methods.

