# OpenReview forum: "PredNext: Explicit Cross-View Temporal Prediction for Unsupervised Learning in Spiking Neural Networks"
_ICLR.cc/2026/Conference — ICLR 2026 Poster_

### Official Review · Reviewer_BJ1t · 2025-10-23

**Soundness:** 2
**Presentation:** 3
**Contribution:** 3
**Rating:** 4
**Confidence:** 4

**Summary:**

This paper proposes PredNext, a novel self-supervised learning framework for deep Spiking Neural Networks (SNNs) on video data. PredNext explicitly models temporal relationships through cross-view future feature prediction, incorporating both step-wise and clip-level prediction mechanisms to enhance semantic consistency and improve feature generalization. Designed as a modular plug-and-play component, PredNext can be seamlessly integrated with a variety of existing self-supervised methods (e.g., SimCLR, BYOL, SimSiam) without modifying backbone architectures. The authors further establish the first systematic benchmark for self-supervised SNNs on large-scale video datasets, including UCF101, HMDB51, and MiniKinetics. Experimental results demonstrate that PredNext consistently yields substantial performance gains and achieves performance comparable to supervised ImageNet pretraining, enabling effective unsupervised temporal representation learning for deep SNNs on complex video tasks.

**Strengths:**

1. Traditional SNNs rely solely on the autoregressive spatiotemporal dynamics of LIF neurons to capture temporal information, which recent studies have suggested is insufficient. This paper provides new insights into this issue, which is highly commendable.

2. This work contributes valuable and important baselines by adapting several classical unsupervised learning algorithms to SNNs. This represents a meaningful contribution to the community.

3. The authors conducted extensive experiments that convincingly demonstrate the effectiveness of the proposed method.

**Weaknesses:**

1. It is recommended to revise the introductory description around line 79. Specifically, the authors should emphasize that the temporal processing capability of SNNs originates from the intrinsic dynamics of spiking neurons rather than solely from the membrane potential. Moreover, not all neurons produce linear outputs below the threshold, so the current wording may be misleading.

2. Figure 2 appears overly complex and does not leverage color cues to guide the reader’s attention, which may hinder comprehension of the algorithm. It is very likely that readers will find this figure difficult to understand by the end of the Introduction. This issue may be alleviated by briefly explaining the roles of  $P_T$ and $P_C$ either within the figure or in close proximity to it.

3. Some items in Table 1 could be formatted in bold to highlight key information more effectively.

4. In Table 2, it is suggested to use checkmarks and crosses in a consistent font style or ensure they originate from the same LaTeX package for visual uniformity.

**Questions:**

1. In Line 3 of Algorithm 1, it appears that the notation suggests $x_i$=$x_j$. If the authors intend to express that the outputs of the data augmentation function differ due to randomness applied at different times, the current formulation may be misleading and could benefit from clarification.

2. In Line 4 of the algorithm, the symbol T seems to be used to represent the number of time steps. However, if I understand correctly, throughout the paper T denotes data augmentations (or is associated with the augmentation function), while N should represent the temporal length. Similar symbol inconsistencies appear in several places. If my interpretation is incorrect, I would appreciate the authors’ clarification.

3. Could the authors provide a theoretical analysis to support the problem formulation and the effectiveness of PredNext, instead of relying solely on empirical observations? Even a supplementary theoretical justification in the appendix would significantly strengthen the submission, especially considering ICLR’s expectations.

4. Could the authors more clearly emphasize the direct relationship between this work and SNNs? The current presentation seems to focus more on modeling spatiotemporal features between frames, rather than leveraging the intrinsic autoregressive temporal dynamics of spiking neurons within frames, which is typically a core property of SNNs.

5. The authors claim that previous work did not train unsupervised SNNs to deeper network depths. However, this is not entirely accurate, and the proposed method also does not extend beyond SEW-ResNet18 in depth. Do the authors have additional evidence or analysis supporting this claim? In particular, a statistical comparison of unsupervised methods applied to deeper SNNs or larger parameter scales would be crucial to validate the novelty emphasized in the manuscript.

---

> ### Author Response · Authors · 2025-11-21
> **Response to Reviewer BJ1t (I / IV)**
>
> We sincerely thank the reviewer for the constructive feedback on our manuscript. The reviewer's comments have enhanced the clarity, rigor, and presentation quality of our work. Following the your's suggestions, we have provided point-by-point responses addressing `manuscript presentation`, `clarity of mathematical notation`, `additional analyses`, and `experiments with deeper networks`:
>
> 1. **It is recommended to revise the introductory description around line 79. Specifically, the authors should emphasize that the temporal processing capability of SNNs originates from the intrinsic dynamics of spiking neurons rather than solely from the membrane potential. Moreover, not all neurons produce linear outputs below the threshold, so the current wording may be misleading.**
>
>     **Reply 1:** We thank the reviewer for the suggestion. We acknowledge the lack of clarity of this sentence in our original expression and have reorganized the language following the your's recommendation:
>
>     *"The temporal processing capability of spiking neural networks stems from the intrinsic dynamics of spiking neurons, which serve as information carriers across timesteps. Standard LIF neurons accumulate membrane potential to retain temporal information and emit discrete spikes when the potential exceeds a threshold."*
>
>     We have incorporated this revised content into the manuscript.
>
> 2. **Figure 2 appears overly complex and does not leverage color cues to guide the reader’s attention, which may hinder comprehension of the algorithm. It is very likely that readers will find this figure difficult to understand by the end of the Introduction. This issue may be alleviated by briefly explaining the roles of $P_T$ and $P_C$ either within the figure or in close proximity to it.**
>
>     **Reply 2:** We thank the reviewer for the suggestion. We have updated the figure and the caption to more clearly illustrate the content of our diagrams. We use red and green to represent the two prediction pathways respectively, with annotations indicating $P_T$ and $P_C$.
>
> 3. **Some items in Table 1 could be formatted in bold to highlight key information more effectively.**
>
>     **Reply 3:** We thank the reviewer for the suggestion. We have highlighted key information following the recommendation to enhance readability.
>
> 4. **In Table 2, it is suggested to use checkmarks and crosses in a consistent font style or ensure they originate from the same LaTeX package for visual uniformity.**
>
>     **Reply 4:** We thank the reviewer for the suggestion. We have revised the checkmarks and crosses in Table 2 following the recommendation, uniformly using `\ding{51}` and `\ding{55}` from the same package to ensure consistency.
>
> 5. **In Line 3 of Algorithm 1, it appears that the notation suggests $x_i$=$x_j$. If the authors intend to express that the outputs of the data augmentation function differ due to randomness applied at different times, the current formulation may be misleading and could benefit from clarification.**
>
>     **Reply 5:** We thank the reviewer for the suggestion. We agree that clearer expression is needed here. We intend to convey that the same sampled instance undergoes data augmentation to produce different augmented samples. We have revised the notation to $x_i=H(x), x_j = H(x)$ for clarity. Following Question 6, we have unified the notation by consistently denoting data augmentation operations as $H$ and reserving $T$ exclusively for timesteps.
>
> 6. **In Line 4 of the algorithm, the symbol T seems to be used to represent the number of time steps. However, if I understand correctly, throughout the paper T denotes data augmentations (or is associated with the augmentation function), while N should represent the temporal length. Similar symbol inconsistencies appear in several places. If my interpretation is incorrect, I would appreciate the authors’ clarification.**
>
>     **Reply 6:** We sincerely thank the reviewer for identifying the symbol confusion issue. We have unified the notation throughout the manuscript, where $T$ denotes the number of timesteps, $H$ represents data augmentation operations, and $N$ typically indicates batch size.

---

> ### Author Response · Authors · 2025-11-21
> **Response to Reviewer BJ1t (II / IV)**
>
> 7. **Could the authors provide a theoretical analysis to support the problem formulation and the effectiveness of PredNext, instead of relying solely on empirical observations? Even a supplementary theoretical justification in the appendix would significantly strengthen the submission, especially considering ICLR’s expectations.**
>
>     **Reply 7:** We sincerely thank the reviewer for the suggestion regarding theoretical analysis of PredNext's effectiveness. We explain from an information-theoretic perspective why predictive modeling naturally enhances temporal consistency and semantic richness of learned representations.
>
>     In the original method, computation focuses on modeling relationships between sample instances. In this work, we further attend to computational interactions between frames and clips, which are unique characteristics of temporal data.
>     Video data contains two types of information: (1) **semantic content** $\mathcal{S}$ such as action categories and object identities, which remains relatively stable over time; (2) **low-level noise** $\mathcal{N}$ such as illumination variations and camera shake, whose temporal correlation decays rapidly. These two information types exhibit fundamentally different temporal correlation characteristics[1,3]: semantic content demonstrates long-range correlation $\rho_{\mathcal{S}}(m) \approx e^{-\epsilon_{\mathcal{S}} m}$, while noise exhibits exponential decay $\rho_{\mathcal{N}}(m) \approx e^{-\lambda_{\mathcal{N}} \cdot m}$[4], where $\epsilon_{\mathcal{S}} \ll \lambda_{\mathcal{N}}$. This implies that a "sport action" persists across multiple frames, whereas "instantaneous glare at a particular moment" quickly disappears.
>
>     PredNext's temporal prediction objective $\mathcal{L} p$ is equivalent to maximizing mutual information $I(z^t; z^{t+m})$ or $I( z; z^* )$ . $z^*$ denotes the temporally aggregated representation of next clip. Assuming semantic and noise statistics are approximately independent. This assumption is generally reasonable for video data, as short-term noise and long-term semantics occupy separated signal frequency spectra[2]: $\mathcal{Z} = \rho_{\mathcal S}(m) + \rho_{\mathcal S}(m)$. For prediction step $m$, the noise mutual information $I(n^t; n^{t+m}) \propto e^{-2\lambda_{\mathcal{N}}m}$ approaches zero, while the semantic mutual information $I(s^t; s^{t+m})$ remains substantial. Consequently, the optimization process naturally prioritizes encoding predictable semantic content while filtering unpredictable noise. Predictability serves as an implicit regularizer that filters out unpredictable noise. This also explains why enforced consistency proves detrimental: the forced constraint $\mathcal{L}_ f = \mathbb{E}_{i,t,s} [1 - \cos( f_i^t, f_i^s )]$ indiscriminately suppresses all temporal variations.
>
> 8. **Could the authors more clearly emphasize the direct relationship between this work and SNNs? The current presentation seems to focus more on modeling spatiotemporal features between frames, rather than leveraging the intrinsic autoregressive temporal dynamics of spiking neurons within frames, which is typically a core property of SNNs.**
>
>     **Reply 8:** We sincerely appreciate the reviewer's suggestion regarding clarifying the relationship between this work and SNNs. We agree that explicitly stating this relationship enhances the clarity of our manuscript. The temporal dynamics in spiking neurons are crucial for the entire network. However, we argue that solely relying on neuronal dynamics to implicitly learn temporal characteristics does not fully exploit the potential of spiking neurons. On one hand, SNN architectures typically borrow from ANN image recognition network designs, which makes networks more prone to spatial bias. Similar observations have been made in ANN-based video models[3,5]. On the other hand, SNNs lack the progressive temporal aggregation mechanisms present in ANN 3D[6] convolutional networks, preventing temporal dimensions from undergoing gradual downsampling through pooling layers or larger-stride convolutions as spatial dimensions do, thereby limiting sufficient temporal information extraction. Therefore, we aim to explicitly enhance temporal consistency through architectural design, thereby alleviating the network's spatial bias while improving temporal aggregation capability to more fully leverage the temporal processing capacity of spiking neurons. We have incorporated this explanation into the revised manuscript.

---

> ### Author Response · Authors · 2025-11-21
> **Response to Reviewer BJ1t (III / IV)**
>
> 9. **The authors claim that previous work did not train unsupervised SNNs to deeper network depths. However, this is not entirely accurate, and the proposed method also does not extend beyond SEW-ResNet18 in depth. Do the authors have additional evidence or analysis supporting this claim? In particular, a statistical comparison of unsupervised methods applied to deeper SNNs or larger parameter scales would be crucial to validate the novelty emphasized in the manuscript.**
>
>     **Reply 9:** We thank the reviewer for the suggestion regarding experiments with deeper SNNs. We agree that incorporating deeper network experiments enhances the completeness of our manuscript. We emphasize that our focus lies in unsupervised training methods on larger-scale temporally rich datasets, as the temporal characteristics of SNNs are difficult to manifest on static images and commonly used small-scale DVS datasets as mentioned in the manuscript. Therefore, we employ larger-scale video datasets for training.
>
>     Furthermore, as shown in `Table 1`, we presents performance across different methods on UCF101. We observe that model performance correlates significantly with pretrained weight effectiveness. PredNext without ImageNet pretraining even outperforms ANN-based supervised baselines (we find that weak augmentation causes ANN collapse on UCF101, thus we employ stronger augmentation than reported). When using ImageNet pretrained weights, PredNext's performance is limited by lower SNN pretrained weight quality `(63.2%[7] vs. 73.2%[8])` compared to methods with larger parameters and ANN supervision. Meanwhile, PredNext performance scales with model size, improving from SEW-ResNet-18 to ResNet-34. However, on SEW-ResNet-50, marginal differences in pretrained weight quality prevent further leveraging parameter scale advantages. Notably, PredNext without pretraining surpasses self-supervised ANN methods with identical architecture (ResNet-18), demonstrating significant advantages in advancing SNN self-supervised learning performance. We have incorporated this content into the revised manuscript.
>
>     | **method**           | **Un-/Sup**                 | **model**                      | **pretrain** |      **pretrain Acc in ImageNet**   | **Top1** | **Top5** |
>     |----------------------|-----------------------------|--------------------------------|--------------|:-----------------------------------:|:--------:|----------|
>     | vanilla              | supervised                  | ResNet 18(ANN)                 |       ✗      |                  -                  |   40.7   |   63.8   |
>     | vanilla              | supervised*                 | ResNet 18(ANN)                 |       ✗      |                  -                  |   53.2   |   78.3   |
>     | vanilla              | supervised*                 | ResNet 34(ANN)                 |       ✗      |                  -                  |   54.2   |   77.4   |
>     | vanilla              | supervised*                 | ResNet 50(ANN)                 |       ✗      |                  -                  |   54.3   |   77.5   |
>     | ReSpike          | supervised                  | ResNet 18(ANN) +MS-ResNet18    |       ✓      |                 73.2                |   77.5   |   93.9   |
>     | SVFormer-st     | supervised*                 | SVFormer-st                    |       ✓      |                 82.9                |   80.2   |     -    |
>     | LSM+STDP          | hand-crafted +supervised    | LSM-16.2M                      |       -      |                  -                  |   70.2†  |     -    |
>     | STS ResNet       | supervised                  | STS ResNet                     |       ✗      |                  -                  |   42.1   |     -    |
>     | $SimSiam$            | unsupervised                | ResNet 18(ANN)                 |       ✗      |                  -                  |   49.3   |   78.6   |
>     | $SimSiam_{PredNext}$ | unsupervised                | SEW ResNet18                   |       ✗      |                  -                  |   54.9   |   82.8   |
>     | $SimCLR_{PredNext}$  | unsupervised                | SEW ResNet18                   |       ✗      |                  -                  |   59.5   |   85.3   |
>     | $SimSiam_{PredNext}$ | unsupervised                | SEW ResNet18                   |       ✓      |                 63.2                |   72.2   |   91.8   |
>     | $SimSiam_{PredNext}$ | unsupervised                | SEW ResNet34                   |       ✓      |                 67.0                |   74.1   |   93.1   |
>     | $SimSiam_{PredNext}$ | unsupervised                | SEW ResNet50                   |       ✓      |                 67.8                |   74.2   |   93.1   |
>
>     Table 2：Comparison with other SNN/ANN methods.

---

> ### Author Response · Authors · 2025-11-21
> **Response to Reviewer BJ1t (IV / IV)**
>
> [1] Taylor G W, Fergus R, LeCun Y, et al. Convolutional learning of spatio-temporal features[C]//European conference on computer vision. Berlin, Heidelberg: Springer Berlin Heidelberg, 2010: 140-153.
>
> [2] Ruderman D, Bialek W. Statistics of natural images: Scaling in the woods[J]. Advances in neural information processing systems, 1993, 6.
>
> [3] Goyal R, Ebrahimi Kahou S, Michalski V, et al. The" something something" video database for learning and evaluating visual common sense[C]//Proceedings of the IEEE international conference on computer vision. 2017: 5842-5850.
>
> [4] Wiskott L, Sejnowski T J. Slow feature analysis: Unsupervised learning of invariances[J]. Neural computation, 2002, 14(4): 715-770.
>
> [5] Choi J, Gao C, Messou J C E, et al. Why can't i dance in the mall? learning to mitigate scene bias in action recognition[J]. Advances in Neural Information Processing Systems, 2019, 32.
>
> [6] Carreira J, Zisserman A. Quo vadis, action recognition? a new model and the kinetics dataset[C]//proceedings of the IEEE Conference on Computer Vision and Pattern Recognition. 2017: 6299-6308.
>
> [7] Fang W, Yu Z, Chen Y, et al. Deep residual learning in spiking neural networks[J]. Advances in Neural Information Processing Systems, 2021, 34: 21056-21069.
>
> [8] He K, Zhang X, Ren S, et al. Deep residual learning for image recognition[C]//Proceedings of the IEEE conference on computer vision and pattern recognition. 2016: 770-778.
>
> ---
>
> We once again sincerely thank the reviewer for the constructive suggestions on our manuscript. We hope our responses have adequately addressed the reviewer's concerns and that our revisions can positively contribute to the reviewer's evaluation and assessment of our work.

---

> > ### Comment · Reviewer_BJ1t · 2025-11-23
> > **Very comprehensive response, I will revise my score.**
> >
> > In fact, this paper itself does not have too many issues. The purpose of raising the aforementioned questions was merely to facilitate better presentation of this article. Fortunately, the authors quickly revised the article in one go to make it most suitable for presentation.
> >
> > The authors have done an excellent job, and I will revise my score to 6 points. Best wishes for everything.

---

### Official Review · Reviewer_vPjs · 2025-10-28

**Soundness:** 3
**Presentation:** 3
**Contribution:** 3
**Rating:** 6
**Confidence:** 3

**Summary:**

The paper proposes PredNext, a plug-and-play auxiliary module for unsupervised SNNs that adds two explicit temporal prediction heads: Step Prediction  and Clip Prediction , trained with cross-view targets. The method is integrated into standard SSL frameworks (SimCLR, MoCo, SimSiam, BYOL, Barlow Twins) and evaluated on UCF101, HMDB51, and miniKinetics. PredNext consistently improves SNN baselines on classification and retrieval and reports analyses of “temporal feature consistency,” arguing that explicit prediction improves consistency more productively than forced consistency constraint.

**Strengths:**

1) Even without deep familiarity with DVS video, I could follow the technical pipeline and reasoning; clarity and readability are clear strengths of this submission.

2) The cross-view PT/PC heads are simple and slot into multiple SSL recipes for SNNs. This idea is a practical contribution for the community.

3) The paper moves beyond small DVS benchmarks to UCF101, HMDB51, and miniKinetics, with coherent pretrain/finetune protocols.

4) Across SimCLR/MoCo/SimSiam/BYOL/Barlow Twins, PredNext improves top-1/5 and retrieval R@K; ablations illustrate the impact of PT vs. PC and step length.

5) The paper formalizes a consistency error and contrasts explicit prediction vs. direct consistency penalties, a useful negative result.

**Weaknesses:**

1) Fig. 1(b): The “distribution of video features in high-dimensional space” appears as a schematic; it’s unclear whether it represents t-SNE or merely an illustration. Define what **blue vs. red** denote, what **green arrows** represent, and  what the inter and intra- cluster distance encodes.  Fig. 2 remains visually cluttered. Arrows cross and re-enter modules, making the flow hard to follow. Please improve the labeling.
2) Please **number** all equations. In the “final optimization objective,” specify whether **α** is fixed across datasets or tuned per dataset. Provide an ablation varying **α** to assess the relative importance of each view’s learning target and report the setting you recommend.
3) **Positioning vs. strong ANN baselines is incomplete.** Table 3 has **no ANN SSL baselines** under the same spatiotemporal setup (e.g., ResNet-18/34  or a Video Transformer). Since the paper argues SNNs better capture long-range temporal dependencies/consistency, please add **ANN-SSL counterparts** trained under the identical clip lengths, strides, and augmentations. This will solidify the “SNN advantage” claim.
4) **Scaling.** How do results change with **SEW-ResNet-34/50** and with a **spiking transformer** backbone?

**Questions:**

Please refer to the four questions included in **Weaknesses** for details.

---

> ### Author Response · Authors · 2025-11-21
> **Response to Reviewer vPjs (I / III)**
>
> We sincerely thank the reviewer for the positive and constructive feedback on our manuscript. The reviewer's comments have significantly enhanced the rigor of our work, making the paper more complete and accessible. Following the reviewer's suggestions, we have made point-by-point revisions addressing: `further clarification of the schematic diagram`, `equation numbering`, `analysis of $\alpha$`, `comparison with ANN baselines`, and `comparison with deeper SNN architectures`. Our detailed point-by-point responses are as follows:
>
>
> 1. **Fig. 1(b): The “distribution of video features in high-dimensional space” appears as a schematic; it’s unclear whether it represents t-SNE or merely an illustration. Define what blue vs. red denote, what green arrows represent, and what the inter and intra- cluster distance encodes. Fig. 2 remains visually cluttered. Arrows cross and re-enter modules, making the flow hard to follow. Please improve the labeling.**
>
>     **Reply 1:** We thank the reviewer for the suggestions to improve the figure illustrations. In Figure 1, panel (b) serves as a schematic diagram illustrating the distribution of video features in high-dimensional space. `Blue` points represent the positions of individual SNN-processed video frames in the high-dimensional feature space, while `red` points indicate cluster centers in nearby feature space regions, demonstrating the overall video or sampled clip locations. `Green` and `red` arrows represent intra-video feature attraction across different frames and inter-video spatial repulsion respectively, thereby achieving higher consistency. Additionally, we have revised Figure 2 by employing color and annotation to emphasize key components and distinguish different parts. We use red and green to represent the two prediction pathways respectively, with annotations indicating $P_C$ and $P_T$. We have incorporated these modifications into the revised manuscript.
>
>     We have revised the caption of Figure 1 to the following:
>     > *Figure 1: Analysis of temporal consistency.（a）Evolution of inter-frame feature similarity during SNN training.（b）Distribution of video features in high-dimensional space, demonstrating more concentrated clustering for high-consistency temporal representations. **Blue** points represent features from different timesteps of the same video, while **red** points indicate cluster centers in nearby feature space locations. **Green** and **red** arrows denote intra-video feature attraction across frames and inter-video feature repulsion respectively*
>
> 2. **(a) Please number all equations.**
>
>     **Reply 2(a):** We sincerely thank the reviewer. Following the reviewer's suggestion, we have added numbering to all equations, making our manuscript more clear and readable.
>
>     **(b) In the “final optimization objective,” specify whether α is fixed across datasets or tuned per dataset. Provide an ablation varying α to assess the relative importance of each view’s learning target and report the setting you recommend.**
>
>     **Reply 2(b):** We sincerely thank the reviewer. We agree that analysis of $\alpha$ substantially contributes to the completeness of our work. Consequently, we conducted ablation studies on PredNext under SimSiam and SimCLR frameworks on UCF101 and HMDB51 to demonstrate how feature extraction capability varies with $\alpha$. As shown in `Table 1`, we observe that increasing $\alpha$ to 0.5 progressively enhances feature extraction capability through PredNext. However, when $\alpha$ becomes excessively large($\alpha$=0.8), it suppresses the original self-supervised learning process, ultimately converging to the crossview-only prediction scenario at $\alpha=1$. Furthermore, we observe that $\alpha=0.5$ achieves optimal feature extraction capability. Therefore, we adopt $\alpha=0.5$ as the default configuration throughout all experiments.
>
>     | methods              | dataset |   0  |  0.2 |  0.4 |  0.5 |  0.6 |  0.8 |  1.0 |
>     |----------------------|---------|:----:|:----:|:----:|:----:|:----:|:----:|:----:|
>     | $SimSiam_{PredNext}$ | UCF101  | 50.8 | 52.4 | 53.4 | 54.9 | 53.8 | 52.2 | 52.4 |
>     | $SimCLR_{PredNext}$  | UCF101  | 57.0 | 57.4 | 57.9 | 59.5 | 58.6 | 55.5 | 52.4 |
>     | $SimSiam_{PredNext}$ | HMDB51  | 28.1 | 28.3 | 28.9 | 30.0 | 29.4 | 29.5 | 29.4 |
>
>     Table 1：Ablation results under different weighting coefficients $\alpha$.

---

> ### Author Response · Authors · 2025-11-21
> **Response to Reviewer vPjs (II / III)**
>
> 3. **(a) Positioning vs. strong ANN baselines is incomplete. Table 3 has no ANN SSL baselines under the same spatiotemporal setup (e.g., ResNet-18/34 or a Video Transformer).**
>
>     **Reply 3(a):** We sincerely appreciate the reviewer's suggestion. We agree that incorporating ANN baseline comparisons and analysis enhances the comprehensiveness of our work. Consequently, we have added comparisons with ANN-based ResNet-18/34 and the hybrid model ReSpike. Regarding Transformer architectures, their training requires substantially larger datasets. Even video datasets that are considerably larger than DVS datasets prove insufficient for effective Transformer training within our computational resources, and we therefore consider this as future work. As shown in `Table 2`, we observe that SNN-based implementations exhibit some performance degradation compared to ANN-based methods[1]. We attribute this to two primary factors: ***(i)*** Datasets such as UCF101/HMDB51 require well-trained ImageNet pretrained models to provide image priors during training. However, current SNN models like SEWResNet exhibit significant performance gaps on ImageNet compared to equivalent ANN architectures `(63.2%[2] vs. 73.2%[3])`. Pretrained weight quality substantially impacts downstream model performance. ***(ii)*** The discrete information communication inherent to SNNs. In ANNs, inter-layer communication uses floating-point values, with each video frame undergoing complete forward propagation. Unlike image recognition tasks where multiple timesteps process the same static image, video tasks in SNNs allocate each timestep to individual frames in the temporal sequence. Consequently, each frame is handled by only one timestep, leading to overall performance degradation on video tasks. Nevertheless, our experimental results demonstrate that PredNext further enhances SNN performance, progressively approaching ANN-based model, and PredNext can slightly surpass supervised ANN-based vanilla model without pretraining weight. We have incorporated the comparative analysis between ANN, ReSpike, and our method into the revised manuscript.
>
>     | **method**           | **Un-/Sup**                 | **model**                      | **pretrain** |      **pretrain Acc in ImageNet**   | **Top1** | **Top5** |
>     |----------------------|-----------------------------|--------------------------------|--------------|:-----------------------------------:|:--------:|----------|
>     | vanilla              | supervised                  | ResNet 18(ANN)                 |       ✗      |                  -                  |   40.7   |   63.8   |
>     | vanilla              | supervised*                 | ResNet 18(ANN)                 |       ✗      |                  -                  |   53.2   |   78.3   |
>     | vanilla              | supervised*                 | ResNet 34(ANN)                 |       ✗      |                  -                  |   54.2   |   77.4   |
>     | vanilla              | supervised*                 | ResNet 50(ANN)                 |       ✗      |                  -                  |   54.3   |   77.5   |
>     | ReSpike           | supervised                  | ResNet 18(ANN) +MS-ResNet18    |       ✓      |                 73.2                |   77.5   |   93.9   |
>     | SVFormer-st      | supervised*                 | SVFormer-st                    |       ✓      |                 82.9                |   80.2   |     -    |
>     | LSM+STDP         | hand-crafted +supervised    | LSM-16.2M                      |       -      |                  -                  |   70.2†  |     -    |
>     | STS ResNet        | supervised                  | STS ResNet                     |       ✗      |                  -                  |   42.1   |     -    |
>     | $SimSiam$            | unsupervised                | ResNet 18(ANN)                 |       ✗      |                  -                  |   49.3   |   78.6   |
>     | $SimSiam_{PredNext}$ | unsupervised                | SEW ResNet18                   |       ✗      |                  -                  |   54.9   |   82.8   |
>     | $SimCLR_{PredNext}$  | unsupervised                | SEW ResNet18                   |       ✗      |                  -                  |   59.5   |   85.3   |
>     | $SimSiam_{PredNext}$ | unsupervised                | SEW ResNet18                   |       ✓      |                 63.2                |   72.2   |   91.8   |
>     | $SimSiam_{PredNext}$ | unsupervised                | SEW ResNet34                   |       ✓      |                 67.0                |   74.1   |   93.1   |
>     | $SimSiam_{PredNext}$ | unsupervised                | SEW ResNet50                   |       ✓      |                 67.8                |   74.2   |   93.1   |
>
>     Table 2：Comparison with other SNN/ANN methods.

---

> ### Author Response · Authors · 2025-11-21
> **Response to Reviewer vPjs (III / III)**
>
> 3. **(b) Since the paper argues SNNs better capture long-range temporal dependencies/consistency, please add ANN-SSL counterparts trained under the identical clip lengths, strides, and augmentations. This will solidify the “SNN advantage” claim.**
>
>     **Reply 3(b):** We sincerely appreciate the reviewer's suggestion. However, we need to clarify that in the manuscript, we intend to convey that vanilla SNN models relying solely on intrinsic temporal transmission through membrane potential fail to achieve superior temporal modeling and thus cannot capture long-term dependencies and consistency, as stated in the Introduction section. Therefore, we aim to explicitly enable the model to learn temporal characteristics through architectural design, thereby achieving better long-term dependency and consistency compared to vanilla SNNs. We have strengthened this expression in the revised manuscript to avoid confusion. We agree that incorporating corresponding ANN SSL models in the comparison is essential for more intuitive evaluation of our method. We have added ANN-based ResNet-18 as a comparative method under SimSiam. As shown in `Table 2`, under our approach, SNN performance marginally exceeds both supervised ANN and ANN-SSL performance. However, due to the limited effectiveness of SNN pretrained models, our method performs slightly lower than methods like ReSpike that benefit from superior pretrained weights. Results demonstrate that our method helps SNNs achieve better long-term dependencies and consistency, thereby enhancing feature extraction capability.
>
> 4. **Scaling. How do results change with SEW-ResNet-34/50 and with a spiking transformer backbone?**
>
>     **Reply 4:** We sincerely appreciate the reviewer's suggestion. We agree that incorporating larger-scale networks further validates the effectiveness of our method. We conducted experiments with SEW-ResNet-34/50. For the same reasons mentioned previously, Transformer architectures require substantially larger datasets for training, and we therefore consider this as future work. As shown in `Table 2`, we present the performance of SimSiam and SimCLR under PredNext on UCF101. We observe that when equipped with ImageNet pretrained weights, SEW-ResNet-34/50 demonstrate superior generalization compared to SEW-ResNet-18. However, due to dataset limitations, SEW-ResNet-34 and SEW-ResNet-50 exhibit comparable performance. We have incorporated these results into the revised manuscript to enhance its completeness.
>
> [1] Xiao et al, ReSpike: Residual Frames-based Hybrid Spiking Neural Networks for Efficient Action Recognition, Neuromorphic Computing and Engineering (2025)
>
> [2] Fang W, Yu Z, Chen Y, et al. Deep residual learning in spiking neural networks[J]. Advances in Neural Information Processing Systems, 2021, 34: 21056-21069.
>
> [3] He K, Zhang X, Ren S, et al. Deep residual learning for image recognition[C]//Proceedings of the IEEE conference on computer vision and pattern recognition. 2016: 770-778.
>
> ---
>
> We once again sincerely thank the reviewer for the valuable suggestions. Our manuscript has become more complete and rigorous through these revisions. We hope our responses adequately address the reviewer's concerns.

---

> > ### Comment · Reviewer_vPjs · 2025-11-23
> > **Rebuttal addressed my concerns.**
> >
> > I appreciate the authors’ detailed and constructive revisions. The clarifications to Figure 1(b) and the redesign of Figure 2 effectively address my concerns about visual clarity and interpretability. Overall, the revisions substantially improve the manuscript’s clarity, rigor, and completeness. I am satisfied with the authors’ responses and am willing to raise my score to 8.

---

### Official Review · Reviewer_85ng · 2025-10-29

**Soundness:** 3
**Presentation:** 4
**Contribution:** 3
**Rating:** 6
**Confidence:** 4

**Summary:**

The PredNext paper presents a novel module to enhance unsupervised representation learning in Spiking Neural Networks (SNNs), specifically for temporal video data. The core innovation is explicitly modeling temporal relationships by forcing the network to perform cross-view future feature prediction through two heads: Step Prediction and Clip Prediction. PredNext's predictive objective guides the network to learn stable, semantically rich features that are naturally consistent, a superior alternative to merely imposing forced consistency constraints, which the authors show degrades performance. The module ntegrates with existing self-supervised methods, and achieves significant performance gains on large-scale video benchmarks, even matching ImageNet-pretrained supervised weights.

**Strengths:**

The paper presents an original and well-executed contribution to unsupervised learning in spiking neural networks by introducing explicit cross-view temporal prediction to enhance temporal consistency. The idea is conceptually novel within the SNN context, bridging predictive coding principles with modern self-supervised methods like SimSiam and MoCo. The experiments are thorough, spanning multiple large-scale video datasets, and the reported gains are consistent and meaningful. The writing is clear, the methodology is reproducible, and the visualizations effectively support the claims. Overall, the work makes a significant and timely contribution by demonstrating that deep SNNs can achieve competitive unsupervised performance on complex temporal data, pushing the field beyond small-scale, biologically inspired setups.

**Weaknesses:**

The paper’s main weakness is that PredNext mostly improves short-term smoothness rather than learning true long-range temporal structure. Performance drops quickly when predicting beyond one step, showing limited temporal modeling capacity.

The training objective is also sensitive to the loss weight ($\alpha$), but the paper provides no analysis of how this affects stability or generalization. Comparisons are limited to ANN-based self-supervised baselines, leaving out SNN-native or biologically inspired methods such as STDP-based learning. This makes it hard to judge the real advantage of PredNext for spiking systems.

The two extra prediction heads add non-trivial computation during training, which could limit scalability.

Finally, the experiments focus only on visual data and do not test multimodal or event-based inputs, where SNNs typically excel.

**Questions:**

1. The performance appears to peak sharply at one-step prediction ($m=1$). Can the authors clarify whether this is a model limitation or a training stability issue? Have they explored architectures or loss designs that allow for longer temporal prediction without collapse?

2.  Since $\alpha$ directly controls the trade-off between self-supervised and predictive losses, how sensitive is PredNext to its value? Would an adaptive or scheduled weighting improve stability?

3. The paper notes that the prediction heads are only used during training, but could the authors quantify the actual increase in computation and memory footprint? Understanding this cost would help assess scalability to larger networks or datasets.

4. Why were biologically inspired unsupervised methods (e.g., STDP or predictive plasticity-based models) excluded from benchmarking? Including at least one such baseline would better contextualize PredNext’s contribution within SNN research.

5.  PredNext improves temporal consistency but risks over-smoothing. How do the authors ensure that important temporal variations (e.g., motion or action cues) are not suppressed during training?

6. Since SNNs are well-suited to event-driven data, do the authors expect PredNext to transfer effectively to modalities like DVS or multimodal fusion (e.g., vision + optical flow)? If so, how would the framework adapt?

---

> ### Author Response · Authors · 2025-11-21
> **Response to Reviewer 85ng (I / III)**
>
> We sincerely thank the reviewer for the positive evaluation of our manuscript and the valuable suggestions provided. Following the reviewer's recommendations, we have addressed each point regarding `long-term temporal structure`, `analysis of α`, `comparison with STDP methods`, `computational overhead of training heads`, and `other modalities`. The reviewer's suggestions have significantly enhanced the completeness and logical coherence of our paper. Below are our point-by-point responses:
>
> 1. **(W1) The paper’s main weakness is that PredNext mostly improves short-term smoothness rather than learning true long-range temporal structure. Performance drops quickly when predicting beyond one step, showing limited temporal modeling capacity. (Q1) The performance appears to peak sharply at one-step prediction (m=1). Can the authors clarify whether this is a model limitation or a training stability issue? Have they explored architectures or loss designs that allow for longer temporal prediction without collapse?**
>
>     **Reply 1:** We sincerely thank the reviewer for raising the question regarding prediction step size. We apologize for insufficient clarity on this matter and would like to provide the following clarification. PredNext simultaneously incorporates Step prediction and Clip prediction, enabling the model to capture both short-term and long-term temporal structures. Regarding step prediction, when $m>1$, adjacent timesteps lose the ability to interact for prediction, as larger $m$ values cause the model to skip nearby temporal moments, resulting in significantly sparser predictive interactions compared to $m=1$ and consequently leading to performance degradation. To investigate this issue, we explored a cumulative prediction strategy by progressively incorporating $m=2, 3, ...$ on top of the $m=1$ predictions. Experimental results demonstrate that $m=1$ alone provides sufficient temporal modeling capacity without requiring additional prediction steps. We have incorporated this analysis into the revised manuscript.
>
> 2. **(W2a) The training objective is also sensitive to the loss weight ($\alpha$), but the paper provides no analysis of how this affects stability or generalization. (Q2) Since $\alpha$ directly controls the trade-off between self-supervised and predictive losses, how sensitive is PredNext to its value? Would an adaptive or scheduled weighting improve stability?**
>
>     **Reply 2:** We sincerely appreciate the reviewer's suggestion regarding the analysis of $\alpha$. We agree that examining the influence of $\alpha$ on model performance will substantially enhance the completeness of our work. Consequently, we conducted ablation studies on PredNext under SimSiam and SimCLR frameworks on UCF101 and HMDB51 to illustrate how feature extraction capability varies with $\alpha$. As shown in Table 1, we observe that increasing $\alpha$ to 0.5 progressively enhances the model's feature extraction capability through PredNext. However, when $\alpha$ becomes excessively large $(\alpha=0.8)$, it suppresses the original self-supervised learning process, ultimately converging to the crossview-only prediction scenario at $\alpha=1$. Furthermore, we observe that `α = 0.5` achieves the highest feature extraction capability, and this value demonstrates relative stability across different methods. Therefore, we adopt $\alpha=0.5$ as the default configuration throughout all experiments.
>
>     | methods              | dataset |   0  |  0.2 |  0.4 |  0.5 |  0.6 |  0.8 |  1.0 |
>     |----------------------|---------|:----:|:----:|:----:|:----:|:----:|:----:|:----:|
>     | $SimSiam_{PredNext}$ | UCF101  | 50.8 | 52.4 | 53.4 | 54.9 | 53.8 | 52.2 | 52.4 |
>     | $SimCLR_{PredNext}$  | UCF101  | 57.0 | 57.4 | 57.9 | 59.5 | 58.6 | 55.5 | 52.4 |
>     | $SimSiam_{PredNext}$ | HMDB51  | 28.1 | 28.3 | 28.9 | 30.0 | 29.4 | 29.5 | 29.4 |
>
>     Table 1：Ablation results under different weighting coefficients $\alpha$.

---

> ### Author Response · Authors · 2025-11-21
> **Response to Reviewer 85ng (II / III)**
>
> 3. **(W2b) Comparisons are limited to ANN-based self-supervised baselines, leaving out SNN-native or biologically inspired methods such as STDP-based learning. This makes it hard to judge the real advantage of PredNext for spiking systems. (Q4) Why were biologically inspired unsupervised methods (e.g., STDP or predictive plasticity-based models) excluded from benchmarking? Including at least one such baseline would better contextualize PredNext’s contribution within SNN research.**
>
>     **Reply 3:** We thank the reviewer for the suggestion to include comparisons with SNN-native or STDP-based models. We agree that incorporating STDP and biologically inspired methods enhances the comprehensiveness of our comparison, as presented in `Table 2`. However, as demonstrated in the Introduction, models based on biological principles or STDP mechanisms typically struggle with large-scale temporal data processing and cannot perform network-wide optimization as effectively as deep learning models trained via backpropagation. Consequently, STDP or biologically inspired optimization methods generally require specialized architectural designs to meet training requirements and lack the broad applicability of backpropagation-based approaches. We have incorporated LSM with STDP-related methods in the comparison, which employs a non-standard dataset usage protocol, yet we include it to provide a more comprehensive evaluation landscape.
>
>     | **method**           | **Un-/Sup**                 | **model**                      | **pretrain** |      **pretrain Acc in ImageNet**   | **Top1** | **Top5** |
>     |----------------------|-----------------------------|--------------------------------|--------------|:-----------------------------------:|:--------:|----------|
>     | vanilla              | supervised                  | ResNet 18(ANN)                 |       ✗      |                  -                  |   40.7   |   63.8   |
>     | vanilla              | supervised*                 | ResNet 18(ANN)                 |       ✗      |                  -                  |   53.2   |   78.3   |
>     | vanilla              | supervised*                 | ResNet 34(ANN)                 |       ✗      |                  -                  |   54.2   |   77.4   |
>     | vanilla              | supervised*                 | ResNet 50(ANN)                 |       ✗      |                  -                  |   54.3   |   77.5   |
>     | ReSpike           | supervised                  | ResNet 18(ANN) +MS-ResNet18    |       ✓      |                 73.2                |   77.5   |   93.9   |
>     | SVFormer-st      | supervised*                 | SVFormer-st                    |       ✓      |                 82.9                |   80.2   |     -    |
>     | LSM+STDP          | hand-crafted +supervised    | LSM-16.2M                      |       -      |                  -                  |   70.2†  |     -    |
>     | STS ResNet        | supervised                  | STS ResNet                     |       ✗      |                  -                  |   42.1   |     -    |
>     | $SimSiam$            | unsupervised                | ResNet 18(ANN)                 |       ✗      |                  -                  |   49.3   |   78.6   |
>     | $SimSiam_{PredNext}$ | unsupervised                | SEW ResNet18                   |       ✗      |                  -                  |   54.9   |   82.8   |
>     | $SimCLR_{PredNext}$  | unsupervised                | SEW ResNet18                   |       ✗      |                  -                  |   59.5   |   85.3   |
>     | $SimSiam_{PredNext}$ | unsupervised                | SEW ResNet18                   |       ✓      |                 63.2                |   72.2   |   91.8   |
>     | $SimSiam_{PredNext}$ | unsupervised                | SEW ResNet34                   |       ✓      |                 67.0                |   74.1   |   93.1   |
>     | $SimSiam_{PredNext}$ | unsupervised                | SEW ResNet50                   |       ✓      |                 67.8                |   74.2   |   93.1   |
>
>     Table 2：Comparison with other SNN/ANN methods.

---

> ### Author Response · Authors · 2025-11-21
> **Response to Reviewer 85ng (III / III)**
>
> 4. **(W3) The two extra prediction heads add non-trivial computation during training, which could limit scalability. (Q3) The paper notes that the prediction heads are only used during training, but could the authors quantify the actual increase in computation and memory footprint? Understanding this cost would help assess scalability to larger networks or datasets.**
>
>     **Reply 4:** We sincerely appreciate the reviewer's suggestion to analyze the computational cost and memory usage of our method. We agree that this comparison contributes to the comprehensiveness of our manuscript. We have incorporated a table comparing device resource usage during training, as shown in `Table 3`. Our method introduces only marginal increases in training time, GPU memory consumption, and FLOPs. This demonstrates that PredNext maintains low computational overhead while enhancing model performance. We have added this content to the appendix of the revised manuscript.
>
>     |                   |    $SimSiam$  |$SimSiam_{PredNext}$ |     $SimCLR$  | $SimCLR_{PredNext}$|
>     |-------------------|:-------------:|:-------------------:|:-------------:|:------------------:|
>     | **GPU devices**   |       4       |          4          |       4       |          4         |
>     | **Training Time** | 1.39min/epoch |    1.43min/epoch    | 1.20min/epoch |    1.36min/epoch   |
>     | **GPU Memory**    |    12.2G×4    |       12.4G×4       |    12.1G×4    |       12.4G×4      |
>     | **Memory Peak**   |      40GB     |         43GB        |      37GB     |        47GB        |
>     | **FLOPs**         |    1.188G×T   |       1.193G×T      |    1.188G×T   |      1.193G×T      |
>     | **Data Workers**  |       16      |          16         |       16      |         16         |
>     | **Throughput**    | 114.3frames/s |    111.1frames/s    | 132.5frames/s |    116.9frames/s   |
>
>     Table 3：Comparison of device resource usage for unsupervised training under different settings
>
> 5. **(W4) Finally, the experiments focus only on visual data and do not test multimodal or event-based inputs, where SNNs typically excel. (Q6)Since SNNs are well-suited to event-driven data, do the authors expect PredNext to transfer effectively to modalities like DVS or multimodal fusion (e.g., vision + optical flow)? If so, how would the framework adapt?**
>
>     **Reply 5:** We sincerely appreciate the reviewer's suggestion to conduct experiments on DVS datasets. Following this recommendation, we have incorporated results on DVSCIFAR10 and NCALTECH101. As discussed in the manuscript, commonly used DVS datasets typically contain limited data samples and exhibit relatively lower temporal complexity compared to video data. Widely adopted datasets such as DVSCIFAR10 and NCALTECH101 prove challenging for unsupervised training, resulting in less pronounced performance differences. Given the typically short temporal sequences in these datasets, we aggregate frames into larger temporal windows of 64 frames exceeding the standard SNN timestep count, subsequently sampling 10 frames for computation. Regarding optical flow methods, which indeed provide enhanced dynamic characteristics, processing typically employs a two-stream network as the backbone architecture, with the remaining training pipeline following the original training protocol.
>
>     |                      | **DVS-CIFAR10** | **NCALTECH101** |
>     |----------------------|-----------------|-----------------|
>     | $SimSiam$            |       76.3      |       78.5      |
>     | $SimSiam_{PredNext}$ |       76.5      |       78.9      |
>
> Table 4：Comparison of DVS datasets under different settings.
>
> 6. **(Q5) PredNext improves temporal consistency but risks over-smoothing. How do the authors ensure that important temporal variations (e.g., motion or action cues) are not suppressed during training?**
>
>     **Reply 6:** We thank the reviewer for this question. We clarify that PredNext indeed enhances temporal consistency in the network, and results demonstrate that it simultaneously improves feature extraction capability. Through analysis of $\alpha$, which controls the ratio between the original SSL loss and prediction loss, we show that the network typically exhibits an optimal point. When the prediction loss proportion becomes excessively large, it suppresses the network's original SSL learning process. Therefore, we regulate $\alpha$ to prevent over-smoothing issues while maintaining effective temporal modeling.
>
>
> ---
>
> We once again thank the reviewer for the time invested in reviewing our manuscript. The reviewer's suggestions have enhanced the completeness of our work. We hope our responses and the additional analyses adequately address the reviewer's concerns.

---

> > ### Comment · Reviewer_85ng · 2025-11-27
> >
> > The rebuttal covers most of my concerns. The added ablations and the (\alpha) sweep strengthen the paper. The explanation for why m>1 hurts is reasonable, but the paper should state clearly that PredNext mainly handles short- to mid-range temporal structure, not true long-range prediction. The claims here should be toned down.
> >
> > The $\alpha$ results align well with the forced-consistency experiment; linking these two points directly in the paper would make the trade-off easier to follow.
> >
> > The comparison to STDP and SNN-native methods is now more complete, but it should be made explicit that these baselines use different protocols and architectures. This prevents over-interpreting the table.
> >
> > The computational overhead table is clear, and a short summary line in the main text would help readers. The rebuttal mentions new DVS results; these should appear in the appendix with a short note that gains are small due to limited temporal depth. The theoretical explanation in the appendix is helpful, and a brief version belongs in the main text.
> >
> > Overall, the authors addressed the core issues and answered all the questions I had - I would like to raise my score.

---

### Official Review · Reviewer_dmJN · 2025-10-30

**Soundness:** 3
**Presentation:** 3
**Contribution:** 3
**Rating:** 6
**Confidence:** 5

**Summary:**

This paper studies self‑supervised learning (SSL) for spiking neural networks  in video representation learning. The authors propose two pretext tasks tailored to spiking video encoders—Step Prediction and Clip Prediction and systematically benchmark several mainstream SSL paradigms adapted to SNNs on UCF‑101 and HMDB‑51. The work evaluates downstream performance on action classification and video retrieval, and provides an analysis of feature distribution and temporal consistency.

**Strengths:**

1. Presents (to my knowledge) one of the first systematic comparisons of multiple SSL families when ported to SNN video encoders.
2. The Step Prediction and Clip Prediction pretext tasks appear well‑aligned with SNN dynamics. Solid empirical evaluation across two standard video benchmarks with both classification and retrieval. The ablations indicate both pretext components contribute to the gains
3. Analysis on the distribution and temporal consistency of video features in Fig 4 is interesting and helps explain how SSL affects time‑evolving SNN features.
4. Establishing baselines and guidance for SSL with SNNs is useful for the neuromorphic community and may reduce the barrier to entry for future work.

**Weaknesses:**

1. Absolute performance vs. ANN baselines. Your best SNN numbers (e.g., **72.2%** on UCF‑101 and **41.5%** on HMDB‑51) lag a modest but non‑trivial margin behind a supervised **ANN ResNet‑18** initialized from ImageNet (≈**76.3% / 48.9%** as reported in **[1]**). This gap matters for ICLR’s broader audience.
2. Compute, memory, and scalability are under‑specified. SNN-based models scale poorly on modern GPUs due to it's recurrent temporal dimension. Training SNNs over time steps often incurs high **memory and wall‑clock** costs. The paper currently omits **GPU hours** and **peak memory**, for your main settings (e.g., $b=256$, $T=16$). Actionable: Report GPU hours and peak memory per configuration; Add training throughput and inference latency. If efficiency is a motivation, consider reporting sparsity metrics (average spike rate) or proxy energy estimates on commodity hardware.

[1] Xiao et al, ReSpike: Residual Frames-based Hybrid Spiking Neural Networks for Efficient Action Recognition, Neuromorphic Computing and Engineering (2025)

**Questions:**

1. How many GPUs are required to train each main model?
2. While the results of different SSL methods are present in Table 3 and Figure 4, there lack a detailed analysis and explaination on what causes the results to differ. Can you elaborate on the comparison between SSL methods?

---

> ### Author Response · Authors · 2025-11-21
> **Response to Reviewer dmJN (I / III)**
>
> We sincerely thank the reviewer for their positive feedback and recognition of our work. We appreciate the constructive suggestions that have enhanced the logical rigor and completeness of our manuscript. We provide point-by-point responses to the reviewer's comments regarding `comparisons with ANNs`, `computational resource analysis`, and `different SSL method comparisons`:
>
> 1. **Absolute performance vs. ANN baselines. Your best SNN numbers (e.g., 72.2% on UCF‑101 and 41.5% on HMDB‑51) lag a modest but non‑trivial margin behind a supervised ANN ResNet‑18 initialized from ImageNet (≈76.3% / 48.9% as reported in [1]). This gap matters for ICLR’s broader audience.**
>
>     **Reply 1:** We sincerely appreciate your suggestion to compare against ANN-based baselines. We agree with this recommendation and have incorporated ANN-related methods such as ReSpike[1] into our comparisons, as shown in `Table 1` or `Table 9` in the revised manuscript. We observe that SNN-based implementations exhibit some performance degradation compared to ANN-based methods. We attribute this to two primary factors: ***(i)***  Datasets like UCF101/HMDB51 require well-trained ImageNet pretrained models to provide image priors during training. However, current SNN models such as SEWResNet exhibit significant performance gaps on ImageNet compared to equivalent ANN architectures `(63.2%[2] vs. 73.2%[3])`. Pretrained weight quality substantially impacts downstream model performance. ***(ii)*** The discrete information communication inherent to SNNs. In ANNs, inter-layer communication uses floating-point values, with each video frame undergoing complete forward propagation. Unlike image recognition tasks where multiple timesteps process the same static image, video tasks in SNNs allocate each timestep to process individual frames in the temporal sequence. Consequently, each frame is handled by only one timestep, leading to overall performance degradation on video tasks. Nevertheless, our experimental results demonstrate that PredNext further enhances SNN performance, progressively approaching ANN-based model, and PredNext can slightly surpass supervised ANN-based vanilla model without pretraining weight. We have incorporated the comparative analysis between ANN, ReSpike, and our method into the revised manuscript.
>
>
>
>     | **method**           | **Un-/Sup**                 | **model**                      | **pretrain** |      **pretrain Acc in ImageNet**   | **Top1** | **Top5** |
>     |----------------------|-----------------------------|--------|--------------|:---------:|:--------:|----------|
>     | vanilla              | supervised                  | ResNet 18(ANN)                 |       ✗      |                  -                  |   40.7   |   63.8   |
>     | vanilla              | supervised*                 | ResNet 18(ANN)                 |       ✗      |                  -                  |   53.2   |   78.3   |
>     | vanilla              | supervised*                 | ResNet 34(ANN)                 |       ✗      |                  -                  |   54.2   |   77.4   |
>     | vanilla              | supervised*                 | ResNet 50(ANN)                 |       ✗      |                  -                  |   54.3   |   77.5   |
>     | ReSpike           | supervised                  | ResNet 18(ANN) +MS-ResNet18    |       ✓      |                 73.2                |   77.5   |   93.9   |
>     | SVFormer-st       | supervised*                 | SVFormer-st                    |       ✓      |                 82.9                |   80.2   |     -    |
>     | LSM+STDP          | hand-crafted +supervised    | LSM-16.2M                      |       -      |                  -                  |   70.2†  |     -    |
>     | STS ResNet        | supervised                  | STS ResNet                     |       ✗      |                  -                  |   42.1   |     -    |
>     | $SimSiam$            | unsupervised                | ResNet 18(ANN)                 |       ✗      |                  -                  |   49.3   |   78.6   |
>     | $SimSiam_{PredNext}$ | unsupervised                | SEW ResNet18                   |       ✗      |                  -                  |   54.9   |   82.8   |
>     | $SimCLR_{PredNext}$  | unsupervised                | SEW ResNet18                   |       ✗      |                  -                  |   59.5   |   85.3   |
>     | $SimSiam_{PredNext}$ | unsupervised                | SEW ResNet18                   |       ✓      |                 63.2                |   72.2   |   91.8   |
>     | $SimSiam_{PredNext}$ | unsupervised                | SEW ResNet34                   |       ✓      |                 67.0                |   74.1   |   93.1   |
>     | $SimSiam_{PredNext}$ | unsupervised                | SEW ResNet50                   |       ✓      |                 67.8                |   74.2   |   93.1   |
>
>     Table 1：Comparison with other SNN/ANN methods.

---

> ### Author Response · Authors · 2025-11-21
> **Response to Reviewer dmJN (II / III)**
>
> 2. **Compute, memory, and scalability are under‑specified. SNN-based models scale poorly on modern GPUs due to it's recurrent temporal dimension. Training SNNs over time steps often incurs high memory and wall‑clock costs. The paper currently omits GPU hours and peak memory, for your main settings (e.g., b=256, T=16). Actionable: Report GPU hours and peak memory per configuration; Add training throughput and inference latency. If efficiency is a motivation, consider reporting sparsity metrics (average spike rate) or proxy energy estimates on commodity hardware.**
>
>     **Reply 2:** We thank the reviewer for the suggestion regarding computational requirements. We agree to include training device specifications and related metrics in the revised manuscript to better characterize our method, as shown in Table 2. We provide training statistics for PredNext and baselines under batch size 256 and T=16, testing both SimSiam and SimCLR to demonstrate the training process. We maintain identical data augmentation methods to ensure fair comparison, noting that data loading time is negligible compared to model training overhead. As shown in Table 2, we report `GPU count`, `training time` per epoch, `GPU memory` usage, `peak memory` consumption, `equivalent FLOPs` on general-purpose hardware (with lower requirements expected on specialized hardware), `number of data workers`, and `throughput`. We employ multi-GPU training on 4 NVIDIA RTX 4090 GPUs (24GB each). Results demonstrate that PredNext introduces minimal additional computational requirements, completing training on the same hardware configuration as the baseline.
>
>     |                   |    $SimSiam$  |$SimSiam_{PredNext}$ |     $SimCLR$  | $SimCLR_{PredNext}$|
>     |-------------------|:-------------:|:-------------------:|:-------------:|:------------------:|
>     | **GPU devices**   |       4       |          4          |       4       |          4         |
>     | **Training Time** | 1.39min/epoch |    1.43min/epoch    | 1.20min/epoch |    1.36min/epoch   |
>     | **GPU Memory**    |    12.2G×4    |       12.4G×4       |    12.1G×4    |       12.4G×4      |
>     | **Memory Peak**   |      40GB     |         43GB        |      37GB     |        47GB        |
>     | **FLOPs**         |    1.188G×T   |       1.193G×T      |    1.188G×T   |      1.193G×T      |
>     | **Data Workers**  |       16      |          16         |       16      |         16         |
>     | **Throughput**    | 114.3frames/s |    111.1frames/s    | 132.5frames/s |    116.9frames/s   |
>
>     Table 2：Comparison of device resource usage for unsupervised training under different settings
>
> 3. **How many GPUs are required to train each main model?**
>
>     **Reply 3:** Since unsupervised pretraining typically requires a large number of epochs to ensure sufficient model training, and feature extraction capabilities progressively strengthen with increased training epochs, we adopt a substantial epoch count. Additionally, video data processing involves multiple frames, resulting in multiplicatively higher computational costs compared to image recognition datasets. Consequently, we employ multi-GPU training on `4 NVIDIA RTX 4090 GPUs (24GB each)`. Nevertheless, our method remains trainable on 2 or even 1 GPU by adjusting batch size. We have incorporated device usage information, including computation, memory, and training time, into the `Table 2` and revised manuscript.

---

> ### Author Response · Authors · 2025-11-21
> **Response to Reviewer dmJN (III / III)**
>
> 4. **While the results of different SSL methods are present in Table 3 and Figure 4, there lack a detailed analysis and explaination on what causes the results to differ. Can you elaborate on the comparison between SSL methods?**
>
>     **Reply 4:** We thank the reviewer for the question regarding differences among SSL methods. We agree that comparative analysis of SSL methods and their performance variations should be included. In our setting, all five baseline methods are trained on instance discrimination tasks, which represents the most widely studied and adopted pretext task. We observe that SimSiam and MoCo exhibit relatively lower performance compared to the other three methods. We attribute this to the following reasons: SimSiam lacks negative samples compared to other approaches, leading to relatively unstable training, whereas BYOL enhances stability through a momentum encoder. On the other hand, MoCo requires maintaining a memory bank as a negative sample repository, which proves challenging for datasets like UCF101 to sustain a large and consistent bank for effective training. We have incorporated analysis of different methods into the revised manuscript.
>
> [1] Xiao et al, ReSpike: Residual Frames-based Hybrid Spiking Neural Networks for Efficient Action Recognition, Neuromorphic Computing and Engineering (2025)
>
> [2] Fang W, Yu Z, Chen Y, et al. Deep residual learning in spiking neural networks[J]. Advances in Neural Information Processing Systems, 2021, 34: 21056-21069.
>
> [3] He K, Zhang X, Ren S, et al. Deep residual learning for image recognition[C]//Proceedings of the IEEE conference on computer vision and pattern recognition. 2016: 770-778.
>
> ---
> We once again thank the reviewer for the time invested in our manuscript and the constructive suggestions provided. The reviewer's feedback has enhanced the rigor and completeness of our paper. We hope our responses adequately address the reviewer's concerns.

---

> > ### Comment · Reviewer_dmJN · 2025-11-24
> >
> > Thanks for the detailed response; the additional experimental results and analyses you provided have largely addressed my concerns. I keep my initial rating.

---

### Author Response · Authors · 2025-11-21
**Response summary**

We thank all Reviewers, the AC, and the PC for their constructive suggestions on our manuscript. Their feedback has substantially enhanced the completeness, logical coherence, and presentation quality of our work. We have provided point-by-point responses to each reviewer's comments, addressing the following aspects:

(1) **Comparison with ANN vanilla and SSL models.** Following reviewer suggestions, we have added comparisons with ANN supervised methods and SSL approaches to illustrate performance differences between our model and ANNs.

(2) **Analysis of $\alpha$.** We have included ablation studies showing model performance variations under different weighting coefficients $\alpha$ to better characterize model properties.

(3) **Comparison with ANN-based, STDP methods, and DVS datasets.** We have incorporated comparisons with ANN-based works, STDP approaches, and results on DVS datasets to provide more comprehensive evaluation of our model.

(4) **Comparison with deeper models.** We have added performance results for deeper network architectures to demonstrate the scalability of our method.

(5) **System resource usage.** We have included system resource consumption during model training to illustrate computational overhead differences.

(6) **Refinement of manuscript presentation and figures.** We have improved textual descriptions and figure illustrations for clearer presentation of our method.

(7) **Additional model analysis and explanations.** We have incorporated more comprehensive analysis and theoretical explanations to better elucidate the underlying principles of our model.

---

We hope our responses adequately address the reviewers' concerns. We once again thank them for their valuable suggestions and time invested in our manuscript.

---

### Author Response · Authors · 2025-12-02
**Summary of Discussion for ACs, SAC, and PCs**

Dear Reviewers, ACs, SACs and PCs:

We once again thank all reviewers, ACs, SACs, and PCs for their valuable feedback on our manuscript. We have completed **all discussions during the Discussion phase prior to November 27th**.

To assist the AC in decision-making, we would like to provide a brief summary of Discussion phase.

The reviewer comments were of high quality and demonstrated excellent expertise. We have provided point-by-point responses addressing all suggestions, and **all Reviewers(`dmJN`, `85ng`, `vPjs`, `BJ1t`) have confirmed that their concerns have been adequately resolved and provided positive evaluations**. Notably, **Reviewers(`85ng`, `vPjs`, `BJ1t`) explicitly indicated an increase in their score**.

Reviewer **dmJN** **(Initial Score 6-> Final Score 6)** expressed *''the additional experimental results and analyses you provided have largely addressed my concerns.''*

Reviewer **85ng** **(Initial Score 6-> Final Score 8)** expressed *''Overall, the authors addressed the core issues and answered all the questions I had - I would like to raise my score."*

Reviewer **vPjs** **(Initial Score 6-> Final Score 8)** expressed *''I am satisfied with the authors’ responses and am willing to raise my score to 8."*

Reviewer **BJ1t** **(Initial Score 4-> Final Score 6)** expressed *''The authors have done an excellent job, and I will revise my score to 6 points."*

---

We sincerely appreciate the time invested by all reviewers in evaluating our work, and we are pleased to have successfully addressed their concerns.

Sincerely,

*The Authors*

---

### Meta-Review · Area_Chair_kRWQ · 2026-01-02

**Summary:**

## Summary
This paper explores self‑supervised learning (SSL) for spiking neural networks (SNNs) in video representation learning. The paper proposes two pretext tasks tailored to spiking video encoders: Step Prediction and Clip Prediction and systematically benchmark several mainstream SSL paradigms adapted to SNNs on UCF‑101, HMDB‑51, and mini-Kinetics. The work evaluates downstream performance on action classification and video retrieval, and provides an analysis of feature distribution and temporal consistency.

## Review & Decision
The paper receives reviews from 4 expert reviewers. After rebuttal all reviewers are happy with the paper and either raising their score or keep positive opinion about the paper. AC reads all reviews, rebuttal QAs, and the paper and agrees with the reviewers's recommendation, thus recommends accept the paper.

**Reviewer Concerns:**

The main concerns were:

+  Low performance of SNN-based methods compared with ANN-based ones.
+ The sensitive of hyper-parameter $\alpha$.
+ The system resource usage (e.g, GPUs) and throughput.

During rebuttal, the author(s) were able to address most of the concerns.

**Reviewer Scores:**

Initially the scores were 6, 6, 6, 4 and after rebuttal all reviewers are convinced and changed their scores to: 6, 8, 8, 6.

---

### Decision · Program_Chairs · 2026-01-26

Accept (Poster)